# TEMPERATURE-DRIVEN ESCAPE EXPLAINS CRITICAL LEARNING RATES IN ADAPTIVE OPTIMIZATION

## ABSTRACT

The learning rate is a central control parameter in neural network training, even for adaptive optimizers such as Adam, which reduce sensitivity to gradient magnitude differences across parameters. It can either trap models in sharp basin or guide convergence toward flatter regions, yet its selection has largely remained empirical. Here, we introduce a temperature-driven escape framework that provides a heuristic statistical-physics view of second-moment-based learning-rate dynamics. By decomposing Adam's second-moment-normalized mini-batch updates into deterministic drift and stochastic fluctuations, we define an effective temperature $T_{\text{eff}}$ induced by mini-batch noise and apply Kramers' escape theory to derive a closed-form expression for the critical learning rate $\alpha_c$. Experiments on vision tasks (MNIST, CIFAR-10 with MLP, CNN, ResNet) and language tasks (SST-2 with BERT, GPT-2, TinyLlama) show that the theoretically predicted $\alpha_c$ achieves better generalization, whereas far deviations from this critical value lead to degraded performance. Beyond initialization, re-estimating $\alpha_c$ also serves as a qualitative diagnostic tool for probing the nature of minima reached by well-trained models. Our results elevate the learning rate from an empirical hyperparameter to a theoretically principled quantity, providing both a predictive rule for initialization and a new perspective on optimizer dynamics.

## 1 INTRODUCTION

As the global scaling factor of parameter updates, the learning rate determines whether training proceeds smoothly or diverges, while also influencing the optimizer's ability to escape sharp basins and converge to flatter regions Bottou et al. (2018); Yoo et al. (2025); Wilson et al. (2017); Li & Arora (2020); Jastrzębski et al. (2018); Keskar et al. (2017); Roulet et al. (2024); Lewkowycz et al. (2020). When the learning rate is too small, updates become negligible and parameters risk confinement in narrow basins. When it is too large, updates overshoot and destabilize training. Despite its importance, learning-rate selection in practice has largely relied on empirical heuristics such as warmup, cosine decay, or restart schedules Loshchilov & Hutter (2017); Smith (2017); Devlin et al. (2019); Iyer et al. (2023), which perturb the optimizer dynamics in search of suitable basins. This lack of a principled criterion stands in sharp contrast to the fundamental role of the learning rate in shaping both convergence speed and generalization performance.

Most modern deep learning optimizers are based on Adaptive Moment Estimation (Adam) Kingma & Ba (2014) rather than Stochastic Gradient Descent (SGD) Zhang et al. (2020b). While SGD exhibits an implicit bias toward flatter minima (i.e., better generalization) Keskar et al. (2017); Jastrzębski et al. (2018); Chaudhari & Soatto (2018); Yang et al. (2023), its updates are driven by raw gradient magnitudes. This allows high curvature (large gradients) to induce large fluctuations and escape, a mechanism fundamentally different from Adam's updates, which are normalized by second-moment estimates. This raw-gradient treatment limits adaptability across parameter directions and often slows convergence in large-scale networks. In contrast, Adam achieves faster convergence and robustness to gradient scaling, making it the optimizer of choice for training deep neural networks and large language models (LLMs) Loshchilov & Hutter (2019); Dosovitskiy et al. (2021). Intuitively, Adam can be viewed as an *adaptive driver*: it accelerates in flat regions and slows down in sharp basin due to its element-wise normalization of gradients (Fig. 1). Nevertheless, Adam still depends on a global learning rate $\alpha$. When $\alpha$ is too large, the optimizer skips over flat basin; when it is too small, it becomes confined in sharp basin. This leads to a key question: Does there exist

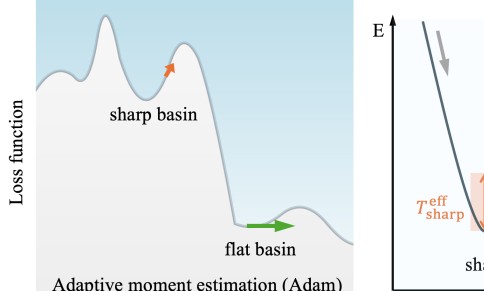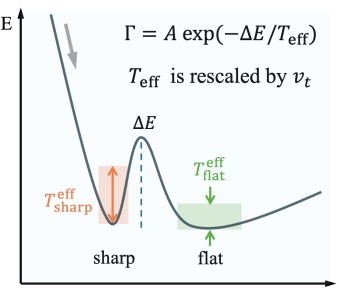

Figure 1: **Temperature-driven escape analysis.** Adaptive moment estimation (Adam) accelerates learning in flat basins while slowing down in sharp basins. Flatter basins are associated with wider parameter changes, meaning better generalization. **Left:** With a small initial global learning rate $\alpha$, the model tends to be trapped in sharp basin due to the reduced effective learning speed (i.e., limited progress under high-curvature regions of the loss landscape). With a large $\alpha$, it may overshoot flat basin since Adam adapts a larger learning speed in basins with small gradients. At the critical learning rate $\alpha \approx \alpha_c$, the optimizer is able to escape sharp basin while preferentially converging to flatter basins. **Right:** Illustration of temperature-driven escape. Here $\theta$ denotes the parameter space, $\Delta E$ is the barrier height between sharp and flat basin. The color bars visualize $T_{\text{eff}}$, the effective temperature induced by mini-batch-induced noise, for the two basins. The escape rate follows Kramers' law, $\Gamma = A \exp(-\Delta E / T_{\text{eff}})$, where $A$ is a constant prefactor reflecting the attempt frequency. Here, $T_{\text{eff}}$ arises from mini-batch-induced noise whose influence is rescaled by the second-moment estimator $v_t$. This coupling amplifies stochasticity in flat basins (small $v_t$) and suppresses it in sharp basins (large $v_t$), enabling curvature-dependent escape. Our goal is to steer the optimizer away from high-curvature regions in early training (gray arrow) by maximizing the relative escape probability.

a critical learning-rate regime in which Adam can escape sharp basins while favoring flatter basins that generalize well?

We address this question by importing concepts from statistical physics. The second-moment normalization in Adam induces a curvature-dependent reshaping of gradient noise, producing two distinct effective temperatures for sharp and flat basins, makes it particularly suitable for temperature-driven escape analysis. To formalize this intuition, we decompose Adam's second-moment-normalized mini-batch updates into deterministic drift and stochastic fluctuations, where gradient noise induces an effective temperature $T_{\text{eff}}$. The escape probability from a basin of barrier height $\Delta E$ then follows Kramers' law Kramers (1940) (Fig. 1),

$$\Gamma = A \exp(-\Delta E / T_{\text{eff}}), \tag{1}$$

where $A$ is a prefactor constant that sets the attempt frequency. Here, the Kramers-style framework is used heuristically and only for a local stability analysis along the dominant positive-curvature direction, quantifying how much the global learning step should provide to overcome second-moment-driven sharpness confinement.

By comparing the effective temperatures associated with sharp and flat basins, with the latter favoring better generalization due to wide basin, this framework leads to a closed-form prediction of the critical learning rate $\alpha_c$. This critical learning rate steers the model away from high-curvature regions early in training, as if the optimizer exactly reaches a sharp minimum, the large accumulation of Adam's second-moment normalization $v_t$ makes escape much harder. Our derivation uses a dominant-eigenmode approximation: escape behavior is modeled along the direction of largest curvature, which empirically governs sharp-basins confinement. Because Adam applies diagonal second-moment normalization, reducing cross-coordinate coupling, this dominant-direction reduction provides a tractable surrogate for the high-dimensional dynamics.

Using temperature-driven escape analysis, the prediction formula provides not only conceptual clarity but also a practical tool for setting the learning rate. Here, we derive a quantitative expression for the critical learning rate $\alpha_c$, determined by the batch size $b$, the mini-batch-induced gradient noise variance $\sigma$, and the dominant Hessian eigenvalue $\lambda_{\max}$ Ghorbani et al. (2019); Sagun et al. (2016). Importantly, $\alpha_c$ can be estimated from only a few mini-batches, enabling principled initialization

without heuristic search or repeated restarts. We validate this prediction across a range of archi-
tectures and datasets: for vision tasks, our formula achieves higher test accuracy on MLPs trained
on MNIST and CNNs/ResNets on CIFAR-10 LeCun et al. (2002); He et al. (2016); for language
tasks, fine-tuning experiments on SST-2 with BERT, GPT-2, and TinyLlama Socher et al. (2013);
Devlin et al. (2019); Radford et al. (2019); Zhang et al. (2024) consistently show that $\alpha_c$ yields better
generalization performance.

In this work, our contributions are as follows:

1. We propose a temperature-driven escape framework that connects second-moment-based
   Adam's optimization dynamics to statistical physics and identifies a critical learning rate
   for escaping sharp basin in the early training and improving generalization.

2. We derive a closed-form initialization rule for the critical learning rate $\alpha_c$, which depends
   only on batch size, gradient noise variance, and the dominant Hessian eigenvalue.

3. We validate the prediction across both vision and language tasks, showing that $\alpha_c$ leads to
   near-optimal performance and better generalization in deep neural networks and LLMs.

4. We show a diagnostic application of $\alpha_c$: re-estimating it during training and resetting the
   learning rate accordingly provides a qualitative probe of whether the model is trapped in
   sharp basin or navigating flatter regions.

## 2 RELATED WORK

**Implicit bias of optimizers.** A central question in understanding optimization is whether certain
algorithms tend to converge to specific regions of the loss landscape. For SGD, extensive work
has shown that its stochasticity induces an implicit bias toward flatter minima, which are associated
with better generalization performance (Hochreiter & Schmidhuber, 1997; Jastrzębski et al., 2018;
Chaudhari et al., 2019; Keskar et al., 2017; Wu et al., 2018). More recent studies have examined
adaptive methods such as RMSProp and AdamW, often noting that while these optimizers con-
verge faster, they may generalize worse than SGD (Loshchilov & Hutter, 2019; Wilson et al., 2017;
Shallue et al., 2019). Unlike SGD, where stochastic noise is directly injected into the parameters,
adaptive optimizers fundamentally reshape the noise distribution via second-moment normalization.
Consequently, existing analyses for adaptive methods remain largely empirical: they rarely provide
a physics-based mechanistic account of how gradient-adaptive updates interact with sharp versus
flat regions of the landscape, nor do they offer principled guidance on how to initialize the global
learning rate.

**Heuristic guidance for Adam.** In practice, Adam remains the optimizer of choice for large-scale
deep networks and language models due to its fast convergence and robustness to gradient scal-
ing (Adam et al., 2015; Devlin et al., 2019; Brown et al., 2020; Dosovitskiy et al., 2021). While
Adam involves several hyperparameters ($\beta_1$, $\beta_2$), prior studies identify the global learning rate $\alpha$
as the most critical factor shaping both convergence and generalization (Adam et al., 2015; Shal-
lue et al., 2019; Bai et al., 2025). Nevertheless, learning-rate selection for Adam is still guided
largely by heuristics such as warmup (Goyal et al., 2017; Devlin et al., 2019; Kalra & Barkeshli,
2024), cosine decay (Loshchilov & Hutter, 2017), or cyclical and restart schedules (Smith, 2017;
Loshchilov & Hutter, 2017). These strategies perturb the optimizer trajectory in search of well-
generalizing basins, but they offer little theoretical understanding and require repeated empirical
tuning. By contrast, our approach provides a closed-form prediction for the critical learning rate,
allowing principled initialization without heuristic search.

**Statistical physics perspectives.** Concepts from statistical physics have been increasingly em-
ployed to analyze optimization and generalization in deep learning (Feng & Tu, 2021; Mandt et al.,
2017; Chaudhari & Soatto, 2018; Jastrzębski et al., 2018; Baldassi et al., 2016; Smith et al., 2021;
Zhu et al., 2019; Gong et al., 2025; Jin et al., 2025; Zhang et al., 2025; Cohen et al., 2022). Prior stud-
ies have modeled SGD dynamics through diffusion approximations (Mandt et al., 2017), Langevin
formulations (Chaudhari & Soatto, 2018), and energy-entropy trade-offs (Smith et al., 2021), pro-
viding insights into noise-induced exploration and the preference for flat minima. However, these
approaches focus predominantly on SGD, leaving adaptive methods far less explored. Our work ad-
vances this line by mapping Adam's gradient-adaptive updates into an effective temperature, which
governs escape from a sharp basin via Kramers' law Kramers (1940).

## 3  PROBLEM SETUP AND METHOD

**Problem setup.**  Prior theoretical analyses of optimization-induced noise predominantly focus on SGD, typically modeling its behavior through diffusion or Langevin dynamics frameworks (Mandt et al., 2017; Chaudhari & Soatto, 2018). However, extending this analysis to adaptive optimizers presents significant challenges. Unlike SGD's uniform step scaling, Adam fundamentally alters the optimization landscape through its gradient-adaptive rescaling mechanism: the running second-moment estimate $v_t$ systematically reduces updates in high-curvature regions while amplifying them in low-curvature areas. This adaptive rescaling replaces the global-uniform step-size scaling used in SGD, thus creating two distinct stochastic regimes. Understanding this sharp–flat asymmetry arising from the second-moment modulation $v_t$ is crucial for predicting how the global learning rate $\alpha$ controls both escape from a sharp basin and convergence toward flatter basins.

**Quadratic approximation of the landscape.**  We analyze Adam's behavior on a smooth loss function $F(\theta)$ with parameters $\theta \in \mathbb{R}^d$, where $d$ is the number of trainable parameters. Near any local minimum $\theta^\star$, we employ the standard quadratic approximation:

$$F(\theta) \approx F(\theta^\star) + \tfrac{1}{2} \sum_{i=1}^{d} \lambda_i (\theta_i - \theta_i^\star)^2, \tag{2}$$

where $\{\lambda_i\}$ represents the Hessian eigenvalues. Here, we use the dominant Hessian eigenvalue $\lambda_{\max}$ as a proxy for local sharpness. We do not require specifying a quantitative threshold that separates sharp and flat minima. Instead, the distinction is purely relative: regions with larger $\lambda_{\max}$ exhibit stronger curvature and empirically lead to poorer generalization, whereas regions with smaller $\lambda_{\max}$ behave as effectively flatter and are associated with better generalization.

**Adam dynamics with stochastic gradients.**  Adam maintains exponential moving averages of both the gradient and its element-wise square. At iteration $t$, the update rules are:

$$m_{t+1} = \beta_1 m_t + (1 - \beta_1) g_t, \tag{3}$$

$$v_{t+1} = \beta_2 v_t + (1 - \beta_2) g_t^2, \tag{4}$$

$$\theta_{t+1} = \theta_t - \alpha \frac{m_{t+1}}{\sqrt{v_{t+1} + \epsilon}}, \tag{5}$$

where $\alpha$ denotes the global learning rate, $\beta_1, \beta_2$ are exponential decay rate constants (typically close to 1), and $\epsilon > 0$ is a small constant for numerical stability. Here, we use a diagonal surrogate for analytical tractability. Because the escape exponent is dominated by the most unstable curvature direction, while the detailed noise covariance affects only the prefactor Langer (1969). Under this surrogate,

$$g_t = \nabla F(\theta_t) + \xi_t, \quad \mathbb{E}[\xi_t] = 0, \quad \mathrm{Var}(\xi_t) = \sigma^2/b, \tag{6}$$

with batch size $b$, where $\sigma^2$ characterizes the variance of the gradient-induced noise.

**Effective temperature derivation.**  To quantify Adam's stochastic behavior, we focus on the random component of parameter updates. For coordinate $i$, the stochastic increment is:

$$\Delta\theta_i^{\mathrm{stoch}} = -\alpha \frac{\delta m_i}{\sqrt{v_i + \epsilon}}, \tag{7}$$

where $\delta m_i$ represents fluctuations in the momentum estimate. Applying the law of total variance under the quasi-stationary assumption for $v_i$:

$$\mathrm{Var}\big[\Delta\theta_i^{\mathrm{stoch}}\big] \approx \alpha^2 \frac{\mathrm{Var}(\delta m_i)}{\langle v_i \rangle}. \tag{8}$$

The momentum variance $\mathrm{Var}(\delta m_i) \approx \sigma^2/b$ inherits directly from the mini-batch gradient noise $\xi_t$, while the expected second moment $\langle v_i \rangle = \mathbb{E}[(\nabla F_i)^2] + \sigma^2/b$ captures both the squared deterministic gradient and the stochastic variance. Detailed derivations are provided in the Appendix.

To establish a bridge between discrete parameter fluctuations and continuous diffusion processes, we invoke the discrete Einstein relation. For a stochastic process with increments $\Delta X$ over time

step $\Delta t$, the variance-diffusion relationship is $\text{Var}[\Delta X] = 2D\,\Delta t$, where $D$ is the diffusion coefficient. Adam's mini-batch processing modifies the effective diffusion time scale. Each algorithmic iteration aggregates gradient information from $b$ samples, condensing the statistical information of $b$ sequential single-sample updates into a single step. This statistical aggregation implies that the natural diffusion time scale should reflect the enhanced efficiency of batch sampling. We therefore define $\Delta t_{\text{eff}} = 1/b$, representing the accelerated information acquisition rate. Substituting into the Einstein relation:

$$D_i^{\text{eff}} = \frac{\text{Var}\big[\Delta\theta_i^{\text{stoch}}\big]}{2\,\Delta t_{\text{eff}}} = \frac{b}{2}\,\text{Var}\big[\Delta\theta_i^{\text{stoch}}\big] \;\approx\; \frac{b\,\alpha^2}{2}\,\frac{\text{Var}[\delta m_i]}{\langle v_i \rangle}. \tag{9}$$

To normalize the influence of batching on the system's stochastic energy, we define the effective temperature as $T_{\text{eff}} = D_{\text{eff}}/\sqrt{\text{Var}(\xi_t)}$, where $D_{\text{eff}}$ denotes the diffusion coefficient induced by Adam's stochastic updates and $\sqrt{\text{Var}(\xi_t)} = \sigma/\sqrt{b}$ is the intrinsic amplitude of the mini-batch-induced noise. This definition ensures that $T_{\text{eff}}$ reflects a *noise-normalized diffusion scale*, quantifying how efficiently stochastic fluctuations are converted into parameter-space exploration rather than merely measuring absolute noise strength. Substituting the expressions for $D_{\text{eff}}$ and $\text{Var}(\xi_t)$ yields

$$T_i^{\text{eff}} = D_i^{\text{eff}}\,\frac{\sqrt{b}}{\sigma_i} \approx \frac{b^{3/2}\alpha^2}{2\sigma_i}\,\frac{\text{Var}[\delta m_i]}{\langle v_i \rangle}, \tag{10}$$

showing that the factor $\sqrt{b}/\sigma_i$ arises directly from normalizing the diffusion coefficient by the intrinsic noise amplitude, rather than from an ad-hoc scaling choice. For SGD, the second-moment normalization term $\langle v_t \rangle$ is effectively constant. This decouples the effective temperature $T_{\text{eff}}$ from the local curvature $\lambda_{\max}$, thereby breaking the self-regulating feedback loop that is central to the derived critical learning rate $\alpha_c$. In the long-time limit where momentum fluctuations are dominated by mini-batch noise, i.e., $\text{Var}[\delta m_i] = \sigma_i^2/b$, yielding:

$$T_i^{\text{eff}} \;\approx\; \frac{\alpha^2 \sigma_i \sqrt{b}}{2\langle v_i \rangle}. \tag{11}$$

**Sharp-flat asymmetry.** The effective temperature exhibits different behavior depending on the local geometry of the loss landscape. In *flat basins*, where gradients are small and curvature is minimal, $\langle v_i \rangle \approx \sigma^2/b$, leading to:

$$T_{\text{flat}}^{\text{eff}} = \frac{\alpha^2 b^{3/2}}{2\sigma}, \tag{12}$$

which exhibits strong stochastic motion.

In *sharp basins*, where the dominant eigenvalue $\lambda_{\max}$ controls the dynamics, a self-consistent analysis yields the cubic relationship $\lambda_{\max}^2 (T^{\text{eff}})^3 = \alpha^2 \sigma b^{1/2}/2$, with solution:

$$T_{\text{sharp}}^{\text{eff}} = \left( \frac{\alpha^2 \sigma b^{1/2}}{2\lambda_{\max}^2} \right)^{1/3}, \tag{13}$$

which noise is strongly suppressed by curvature, making escape far less likely.

**Escape dynamics and critical learning rate.** The effective temperatures directly determine the probability of escaping local basin via Kramers' escape theory Kramers (1940). For a potential barrier of height $\Delta E$, the escape rate follows Eq. 1, and the relative escape probability between flat and sharp basins is given by:

$$R(\alpha) = \frac{\Gamma_{\text{flat}}}{\Gamma_{\text{sharp}}} = \exp\left( -\frac{C_{\text{flat}}}{\alpha^2} + \frac{C_{\text{sharp}}}{\alpha^{2/3}} \right), \tag{14}$$

with

$$C_{\text{flat}} = \frac{2\sigma\Delta E}{b^{3/2}}, \qquad C_{\text{sharp}} = \Delta E \cdot \frac{(2\lambda_{\max}^2)^{1/3}}{b^{1/6}\sigma^{1/3}}. \tag{15}$$

Optimizing this ratio by setting $\frac{dR}{d\alpha} = 0$ yields:

$$\frac{dR}{d\alpha} = R(\alpha)\left( \frac{2C_{\text{flat}}}{\alpha^3} - \frac{2C_{\text{sharp}}}{3\alpha^{5/3}} \right) = 0, \tag{16}$$

leading to the critical learning rate condition $\alpha_c^{4/3} = \frac{3C_{\text{flat}}}{C_{\text{sharp}}}$. We verified this extremum is a maximum by demonstrating that $\frac{d^2R}{d\alpha^2} = R(\alpha)\,\phi''(\alpha) < 0$, where $R(\alpha) > 0$ and the second derivative of the exponent is $\phi''(\alpha_c) = -\frac{8C_{\text{flat}}}{3\alpha_c^4} < 0$ at $\alpha_c$. Solving for $\alpha_c$ explicitly gives the closed-form expression:

$$\alpha_c \approx 3.22\,\frac{\sigma}{b\,\lambda_{\max}^{1/2}}, \tag{17}$$

which provides a principled alternative to heuristic learning rate schedules and enables diagnostic monitoring during training.

**Why the temperature-driven analysis is specific to second-moment adaptive optimizers.** Our derivation of the mini-batch-induced effective temperature, and consequently of the critical learning rate $\alpha_c$, is rooted in the adaptive scaling of the second-moment estimate $v_t$. Specifically, the self-normalization by $v_t$ induces a local cooling effect in high-curvature (sharp) basins. Therefore, the critical (global) learning rate $\alpha_c$ is required to counterbalance this curvature-induced cooling, thereby favoring flatter solutions and better generalization.

## 4 EXPERIMENTS

We evaluate Eq. 17 on vision and language tasks to assess its prediction of the critical learning rate $\alpha_c$. Unless otherwise specified, we follow standard training protocols and vary only the global learning rate $\alpha$, while keeping all other settings fixed: hyperparameters (e.g., $\beta_1 = 0.9$, $\beta_2 = 0.999$), default initialization for vision architectures and preprocessing pipelines. For language tasks (BERT, GPT-2, TinyLlama), we fine-tune from publicly available pretrained checkpoints without modifying weights other than through the chosen learning rate. All experiments were conducted using data center GPUs (NVIDIA A100) and workstation GPUs (NVIDIA RTX A6000).

### 4.1 ESTIMATING THE CRITICAL LEARNING RATE

To operationalize Eq. 17, we estimate the sample-induced gradient noise variance and the dominant Hessian eigenvalue from the first few mini-batches of training. Only early-training curvature is needed because escape is determined in the initial phase, when $v_t$ is still small and the effective temperature permits barrier crossing. As training proceeds, $v_t$ grows sharply in highh-curvature regions and suppresses stochasticity, freezing the dynamics. This early-phase stability of $\lambda_{\max}$ has been observed empirically (Ghorbani et al., 2019; Sagun et al., 2018). The procedure is lightweight and requires only a small number of gradient computations together with a short power iteration, without altering the training pipeline. The estimation proceeds as follows: (*i*) we draw $K$ random mini-batches and compute their gradients $\{g_k\}_{k=1}^{K}$, then use the sample variance of these gradients to estimate $\text{Var}(g)$ and $\sigma$; (*ii*) we estimate the raw $\lambda_{\max}$ by applying $T$ steps of power iteration to the Hessian vector product (HVP) of the loss with respect to the model parameters; and (*iii*) we combine these quantities in the prediction Eq. 17. In practice, we use $K = 20$ mini-batches for variance estimation and $T = 10$ iterations for power iteration, which we found sufficient for stable estimates at negligible cost relative to a single training epoch.

**Computational cost of estimating** $\alpha_c$. In our implementation, we use $K = 20$ gradient samples and $T = 10$ iterations. The first stage requires 20 forward-backward passes, while the second stage costs roughly $1 + T \approx 11$ gradient-equivalent steps (one backpropagation to form the gradient graph and $T$ HVP). Therefore, a single estimate of $(\sigma^2, \lambda_{\max}, \alpha_c)$ costs in total 20 (grad samples) + 1 (graph build) + 10 (HVPs) $\approx 31$ gradient-equivalent steps. Since the FLOPs of a backward pass dominate the training cost of large models, and because both our estimator and standard optimization scale linearly with the number of parameters, their ratio remains unchanged. For comparison, a typical fine-tuning run of a large language model involves $10^4$ to $10^5$ optimization steps Touvron et al. (2023), so the overhead of our estimator is well below $1\%$ of the total training budget, even for billion-parameter models. Crucially, instead of forming the full Hessian, we use HVPs as an efficient alternative, at roughly the cost of a single gradient evaluation, making power iteration practical even at scale.

**Sensitivity of estimating $\alpha_c$.** Table 1 summarizes the sensitivity of our critical learning-rate estimate to perturbations in $\lambda_{\max}$ and $\sigma^2$. Because $\alpha_c \propto (\lambda_{\max})^{-1/2}$, even a 25% error in estimating the top curvature leads to only a modest 10-15% change in $\alpha_c$. In contrast, $\alpha_c$ depends on the noise level as $\alpha_c \propto \sqrt{\sigma^2}$, so a 25% error in $\sigma^2$ results in approximately a 12% change in $\alpha_c$. Overall, even when the curvature or noise estimates may contain uncertainty, the induced error in $\alpha_c$ is considerably smaller, indicating that the critical learning-rate estimate is more stable than either of its raw inputs.

| Estimate error | $\alpha_c$ scaling factor | Relative change (%) |
|---|---|---|
| $\lambda_{\max}+25\%$ | $1.25^{-1/2} \approx 0.894$ | $-10.56\%$ |
| $\lambda_{\max}-25\%$ | $0.75^{-1/2} \approx 1.154$ | $+15.47\%$ |
| $\lambda_{\max}+50\%$ | $1.5^{-1/2} \approx 0.816$ | $-18.35\%$ |
| $\lambda_{\max}-50\%$ | $0.5^{-1/2} \approx 1.414$ | $+41.42\%$ |
| $\sigma^2+25\%$ | $\sqrt{1.25} \approx 1.118$ | $+11.80\%$ |
| $\sigma^2-25\%$ | $\sqrt{0.75} \approx 0.866$ | $-13.40\%$ |
| $\sigma^2+50\%$ | $\sqrt{1.5} \approx 1.225$ | $+22.47\%$ |
| $\sigma^2-50\%$ | $\sqrt{0.5} \approx 0.707$ | $-29.29\%$ |

Table 1: Scaling and relative changes of $\alpha_c$ when $\lambda_{\max}$ or $\sigma^2$ varies by $\pm 25\%$ or $\pm 50\%$.

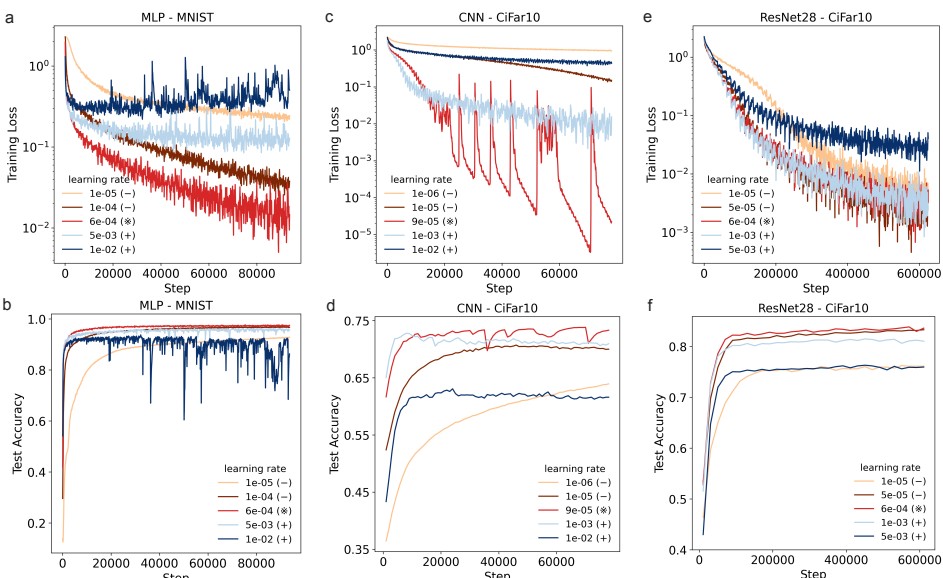

Figure 2: **Training and test performance under varying learning rates across architectures.** (**a**, **b**) MLP on MNIST (batch size 32), (**c**, **d**) CNN on CIFAR-10 (batch size 64), and (**e**, **f**) ResNet-28 on CIFAR-10 (batch size 4). Top row: training loss; bottom row: test accuracy. In each case, the learning rate marked with ※ corresponds to the theoretically predicted $\alpha_c$. Curves labeled (–) denote values smaller than $\alpha_c$, while (+) denotes larger values. Across datasets and architectures, performance peaks near $\alpha_c$, while learning rates that are orders of magnitude smaller or larger either slow down optimization or reduce generalization performance.

## 4.2 IMAGE CLASSIFICATION FROM SCRATCH

We first test on image classification benchmarks. For MNIST, we train a 5-layer MLP with ReLU activations, and for CIFAR-10 we train both a 2-layer CNN and a ResNet-28 from scratch. Figure 2 shows training loss and test accuracy across a range of learning rates. In all settings, learning rates that deviate substantially from the predicted value (marked with ※) lead to degraded performance. Smaller values (marked with –) slow convergence and risk entrapment in narrow basins, while larger values (marked with +) destabilize training and reduce test accuracy. Because Adam is inherently robust to small perturbations of the learning rate, these effects become evident only when the deviation is at least on the order of half a magnitude or more. By contrast, the predicted critical learning rate $\alpha_c$ yields better generalization, even for different batch sizes and loss function (Appendix Fig. 1). Notably, for the CNN trained with larger batch size (64, Fig. 2c,d), runs initialized at $\alpha_c$ exhibit stronger fluctuations, consistent with our theoretical claim that $\alpha_c$ characterizes the transition from confinement in sharp basins to exploration of flatter regions.

### 4.3 FINE-TUNING LANGUAGE MODELS

We next evaluate our prediction in large-scale transformers, by fine-tuning BERT, GPT-2, and TinyLlama-1B models on the SST-2 sentiment classification benchmark. Figure 3 shows both training loss dynamics and peak test accuracy across different choices of initial global learning rate $\alpha$. Across all three models, we observe a consistent pattern of dependence on the learning rate. When $\alpha \ll \alpha_c$, optimization still converges but generalization degrades, consistent with entrapment in narrow basins. Conversely, when $\alpha \gg \alpha_c$, training becomes unstable and test accuracy drops. In contrast, the predicted critical learning rate $\alpha_c$ consistently achieves superior generalization performance, confirming that our framework extends beyond conventional deep networks to transformer-based architectures. In particular, the TinyLlama-1B model shows the largest relative improvement when tuned near $\alpha_c$, suggesting that billion-parameter models not only benefit more from principled learning-rate selection but also that critical-regime dynamics are important at large scale.

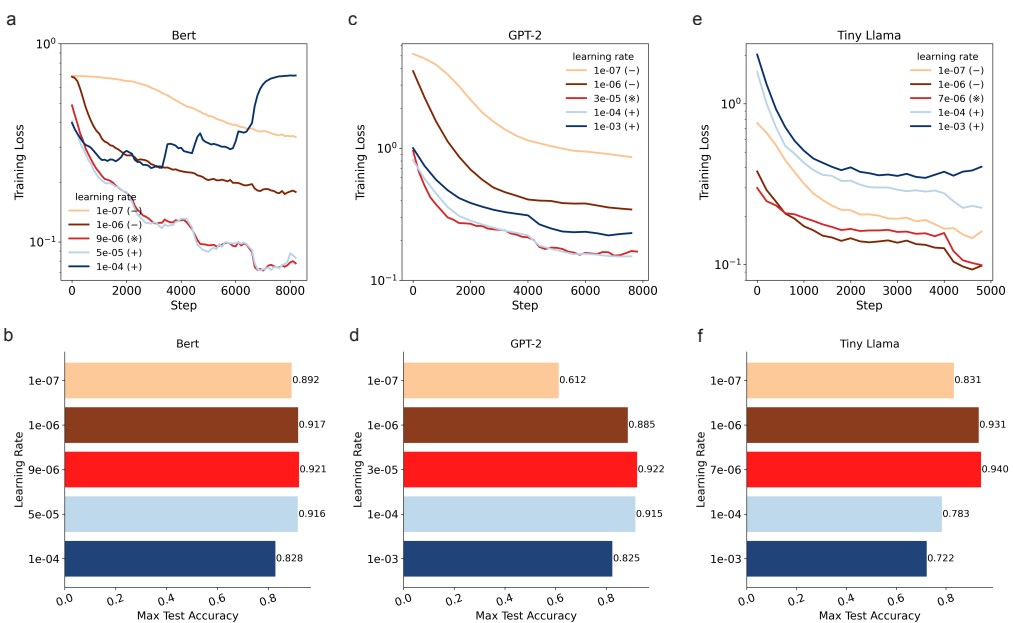

Figure 3: **Fine-tuning pretrained language models on SST-2 under varying learning rates.** (**a**, **b**) BERT, (**c**, **d**) GPT-2, and (**e**, **f**) TinyLlama-1B. Top row: training loss; Bottom row: maximum test accuracy achieved in the training process. In each model, the learning rate marked with (✳) denotes the theoretically predicted critical value $\alpha_c$, while (–) and (+) represent values smaller and larger than $\alpha_c$. All experiments used a batch size of 32. Performance consistently peaks near $\alpha_c$, whereas smaller learning rates slow convergence and reduce accuracy, and larger ones destabilize training.

### 4.4 PERFORMANCE SENSITIVITY OF OPTIMIZERS TO LEARNING RATES

To empirically assess how different optimizers respond to learning rate errors relative to the critical learning rate, we trained TinyLlama-1B (fine-tuned) and ResNet (from scratch) under substantial deviations from the predicted $\alpha_c$. As reported in Table 2, $\alpha_c$ consistently corresponds to a regime of improved generalization across all optimizers, with `warmup+cosine_decay` achieving the highest performance. The additional warmup and cosine decay schedules are particularly effective in mitigating training instability. We also observe a sensitivity pattern: when the learning rate is too small, the model becomes trapped in sharp basins that yield poorer generalization, and when the learning rate is moderately larger, performance remains close to that at $\alpha_c$. However, if the learning rate is too large, the model escapes sharp basins but fails to remain in flatter regions, leading to degraded performance. Moreover, the low standard deviations across independent seeds highlight the intrinsic robustness of second-moment-based optimizers. Comparable trends are observed for ResNet-28 training (Appendix Table 3).

| $\alpha_c \times$ scaling factor ($\pm 25\%$ error) | Adam | AdamW_warmup | AdamW_warmup + cosine_decay | AdamW_cosine_decay | Adafactor |
|---|---|---|---|---|---|
| $\times 10 \pm 25\%$ | $0.860 \pm 0.010$ | $0.845 \pm 0.010$ | $0.875 \pm 0.008$ | $0.885 \pm 0.008$ | $0.820 \pm 0.012$ |
| $\times 5 \pm 25\%$ | $0.925 \pm 0.005$ | $0.915 \pm 0.006$ | $0.935 \pm 0.004$ | $0.938 \pm 0.004$ | $0.910 \pm 0.008$ |
| $\times 2 \pm 25\%$ | $0.935 \pm 0.004$ | $0.938 \pm 0.004$ | $0.945 \pm 0.003$ | $0.942 \pm 0.003$ | $0.935 \pm 0.005$ |
| $\times 1.5 \pm 25\%$ | $0.938 \pm 0.004$ | $0.941 \pm 0.003$ | $0.946 \pm 0.003$ | $\mathbf{0.945} \pm 0.003$ | $0.939 \pm 0.004$ |
| $\times 1 \pm 25\%$ (*) | $\mathbf{0.940} \pm 0.004$ | $\mathbf{0.943} \pm 0.003$ | $\mathbf{0.948} \pm 0.003$ | $\mathbf{0.945} \pm 0.003$ | $\mathbf{0.941} \pm 0.004$ |
| $\times 0.5 \pm 25\%$ | $0.938 \pm 0.004$ | $0.940 \pm 0.004$ | $0.942 \pm 0.003$ | $0.940 \pm 0.004$ | $0.938 \pm 0.005$ |
| $\times 0.1 \pm 25\%$ | $0.925 \pm 0.006$ | $0.928 \pm 0.005$ | $0.930 \pm 0.005$ | $0.928 \pm 0.006$ | $0.920 \pm 0.008$ |

Table 2: Test accuracy (mean $\pm$ s.d.) for TinyLlama-1B on SST-2 across second-moment optimizers. Learning rates include a scaling factor with $\pm 25\%$ error, evaluated over 10 independent runs.

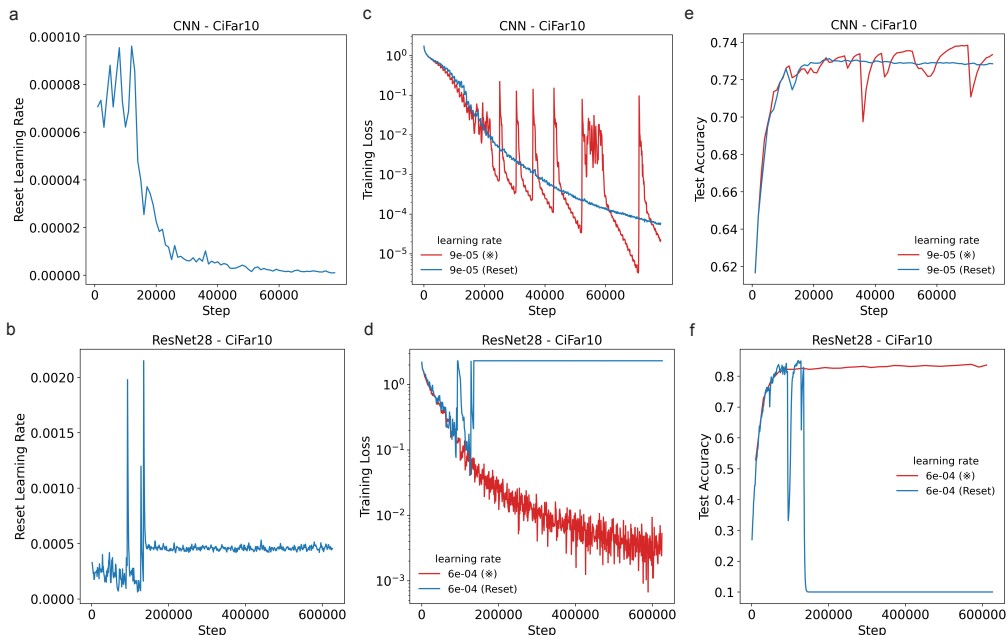

Figure 4: **Re-estimating and resetting the learning rate as a diagnostic probe.** (**a**, **b**) Evolution of the estimated critical learning rate $\alpha_c$ during training for CNN (batch size 64) and ResNet-28 (batch size 4) on CIFAR-10. (**c**, **d**) Training loss when using either a fixed initial learning rate set to the predicted $\alpha_c$ (red, ※) or repeated resets to the locally re-estimated $\alpha_c$ (blue). (**e**, **f**) Corresponding test accuracy. Reset reveals a diagnostic contrast: in a sharp basin, $\alpha_c$ decreases and training remains trapped, while in flatter regions, $\alpha_c$ increases, enabling escape and potential gains in generalization but also introducing the possibility of departing from otherwise stable flat basins.

### 4.5 DIAGNOSTIC USE BEYOND INITIALIZATION

While the critical learning rate $\alpha_c$ is used to guide initialization, it also provides a diagnostic tool for *qualitatively classifying*, rather than *quantifying*, the sharpness of a minimum. To probe the types of minima, we conduct a reset experiment in which the global learning rate $\alpha$ was periodically reset to its current predicted value every $M = 1000$ steps, after which training continued with the default schedule. This intervention revealed two distinct behaviors. First, when the model resided in sharp minima characterized by large local curvature, the reset failed to induce escape and the model remained trapped, with test accuracy stable. Second, when the model was located in flatter basins, resets disrupted the existing solution and in some cases caused the model to collapse, reducing test accuracy to chance level ($\approx 0.1$ for 10 classes). However, for flatter basins, these perturbations also created opportunities to escape local minima and explore alternative basins that supported higher generalization (Fig. 4f), in contrast to the lower generalization associated with sharp minima. This also explains why $\alpha_c$ matters only at the start of training: after $v_t$ accumulates in a sharp basin, escaping would require a learning rate much larger than the initial $\alpha_c$. Moreover, we observe that

large batches bias Adam dynamics toward sharp basins, mirroring prior findings for SGD (Keskar et al., 2017). For example, ResNet-28 trained with a small batch size (4) tended to occupy flatter regions and somehow benefited from resets (Fig. 4d,f), whereas the same model trained with a larger batch size (100, Appendix Fig. 2) became stuck in sharp minima.

### 4.6 BATCH SIZE DEPENDENCE

Furthermore, we investigate the role of batch size $b$. Our theory predicts $\alpha_c \propto b^{-1}$, implying that larger batches could lower the optimal learning rate. However, deviations emerge at the extremes (see Appendix Fig. 3). For very small batches, gradient noise is heavy-tailed and non-Gaussian Simsekli et al. (2019), violating the central limit theorem and forcing stable training to operate at learning rates below the predicted $\alpha_c$. At the other extreme, with very large batches, mini-batch noise vanishes, causing the model to become trapped in a sharp basin unless the learning rate is increased above $\alpha_c$ to compensate for the missing noise. Prior work has shown that SGD becomes deterministic at very large batch sizes (e.g., in the range of thousands) due to diminishing sampling noise Keskar et al. (2017). In contrast, adaptive methods such as Adam suppress gradient noise much more aggressively through second-moment normalization Zhang et al. (2020a), causing the stochasticity to vanish at much smaller batch sizes. These results indicate that the theory is most predictive in the mesoscopic batch-size regime Smith & Le (2017); Jiang et al. (2023), where Gaussian-like noise mediates the balance between curvature effects and sample variance. Moreover, because the escape-rate scaling depends on the curvature-noise balance rather than the precise noise distribution, our predictions hold even when Gaussianity is only approximate.

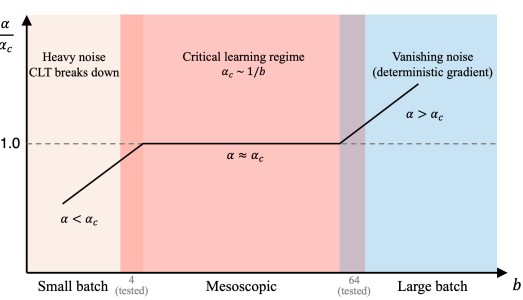

Figure 5: **Batch-size dependence of the critical learning rate.** Schematic illustration of the relationship between the normalized learning rate $\alpha/\alpha_c$ and batch size $b$ in Adam. In the small-batch regime, gradient estimates are dominated by heavy-tailed noise, the central limit theorem (CLT) breaks down, and $\alpha < \alpha_c$ is required to suppress excessive stochasticity. In the mesoscopic regime, a clear scaling $\alpha_c \sim 1/b$ emerges, defining the critical learning regime where noise and curvature balance. In the large-batch regime, the dynamics approach a nearly deterministic gradient descent, and $\alpha > \alpha_c$ is needed to offer the source of noise. Gray markers on the $b$-axis indicate our tested cases.

## 5 DISCUSSION

Our work establishes a statistical-physics interpretation of Adam's learning-rate dynamics by mapping its gradient-adaptive updates to a temperature-driven escape process. This perspective leads to a closed-form initialization guide of the critical learning rate $\alpha_c$, which we show aligns with empirically optimal values across both vision and language tasks, from simple MLPs to billion-parameter transformers. Taken together, these results elevate the learning rate from a heuristic knob to a theoretically grounded quantity that both predicts and interprets generalization in deep learning.

**Limitations and future work.** Despite its effectiveness, our framework rests on several simplifying assumptions. (*i*) We assume Gaussian gradient noise, which breaks down at extremely small batch sizes, where noise distributions are heavy-tailed Simsekli et al. (2019). (*ii*) The rare-escape approximation may fail in large-batch regimes, because Adam accelerates mini-batch noise vanishing by suppressing gradient noise via second-moment normalization. (*iii*) Our critical learning rate-based escape theory does not extend to SGD, as the absence of second-moment normalization yields nearly identical effective temperatures in sharp and flat basins. (*iv*) Future work could extend our dominant-curvature approximation to a full high-dimensional framework that captures valley-like landscapes.

ETHICS STATEMENT

This work is purely theoretical and computational. It does not involve human subjects, personally identifiable information, or sensitive data.

REPRODUCIBILITY STATEMENT

A complete description of the experimental settings, including model architectures, optimization hyperparameters, and training procedures, is provided in the main text and in the Appendix.

THE USE OF LARGE LANGUAGE MODELS

All research ideas, theoretical derivations, experimental design, and analyses were conducted solely by the authors. The use of Large Language Models (LLMs) was limited to minor language refinement, including grammar correction, word choice, and sentence clarity, and it did not influence the scientific content or conclusions.

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

# APPENDIX

**Anonymous authors**

## A  THEORETICAL FRAMEWORK FOR ADAM OPTIMIZER

We consider the Adam optimizer applied to a loss function $F(\theta)$ with mini-batch gradient estimates. The complete dataset is $\mathcal{D} = \{(x_1, y_1), \ldots, (x_N, y_N)\}$, and at each step $t$, we randomly sample a mini-batch $\mathcal{B}_t \subset \mathcal{D}$ of size $b$. Thus, the mini-batch gradient estimate is

$$\nabla f_{\mathcal{B}_t}(\theta_t) = \frac{1}{b} \sum_{(x,y) \in \mathcal{B}_t} \nabla \ell(\theta_t; x, y), \tag{1}$$

where $\ell(\theta; x, y)$ is the per-sample loss function (e.g., cross-entropy for classification) evaluated at parameters $\theta$ and data point $(x, y)$. To analyze the dynamics, the stochastic gradient can be expressed as the sum of the full gradient and a sampling fluctuation

$$\nabla f_{\mathcal{B}_t}(\theta_t) = \nabla F(\theta_t) + \xi_t, \tag{2}$$

where $\nabla F(\theta_t) = \frac{1}{N} \sum_{i=1}^{N} \nabla \ell(\theta_t; x_i, y_i)$ is the full gradient and $\xi_t$ is the mini-batch sampling noise. While mini-batch sampling ensures that gradient noise is unbiased, its covariance in practice is anisotropic and temporally correlated HaoChen et al. (2021); Simsekli et al. (2019); Zhu et al. (2018). For analytical tractability, we approximate the local noise by a diagonal and memoryless surrogate when deriving the effective-temperature scaling. This surrogate does not change the predicted scaling of the critical learning rate: high-dimensional extensions of Kramers theory Langer (1967; 1969) show that the escape exponent is dominated by the most unstable curvature direction, while the full noise covariance affects only the prefactor. Since Adam's per-coordinate second-moment normalization suppresses cross-coordinate correlations, the surrogate preserves the $\alpha_c$ scaling while omitting only prefactor-level effects. Under this surrogate, the gradient noise $\xi_t$ satisfies

$$\mathbb{E}[\xi_{i,t}] = 0, \tag{3}$$

$$\mathrm{Var}[\xi_{i,t}] = \frac{\sigma_i^2}{b}, \tag{4}$$

$$\mathbb{E}[\xi_{i,t}\xi_{j,s}] = \delta_{ij}\,\delta_{ts}\,\frac{\sigma_i^2}{b}, \tag{5}$$

where $\delta_{ij}$ and $\delta_{ts}$ are Kronecker deltas (equal to 1 if the indices are equal and 0 otherwise), and $\sigma_i^2$ denotes the variance of the per-sample gradient in the $i$-th component, i.e.,

$$\sigma_i^2 = \mathbb{E}\left[(\nabla \ell_i(\theta; Z) - \nabla F_i(\theta))^2\right], \tag{6}$$

for a random sample $Z$. These properties arise naturally from the unbiasedness of the mini-batch gradient estimator together with the independence of samples.

To analyze the optimizer's dynamics near local basins, it is essential to characterize the curvature of the loss landscape. A common approach is to approximate the loss function by a quadratic expansion around a local minimum $\theta^*$, where the Hessian matrix $H$ captures the local geometry through its eigenvalues $\{\lambda_i\}$. In the neighborhood of a local minimum $\theta^*$, the loss function can be approximated by

$$F(\theta) = F(\theta^*) + \nabla F(\theta^*)^\top (\theta - \theta^*) + \frac{1}{2}(\theta - \theta^*)^\top H(\theta - \theta^*) + O(\|\theta - \theta^*\|^3), \tag{7}$$

where $H = \nabla^2 F(\theta^*)$ is the Hessian matrix. Since $\theta^*$ is a local minimum, the gradient $\nabla F(\theta^*) = 0$, so

$$F(\theta) \approx F(\theta^*) + \frac{1}{2}(\theta - \theta^*)^\top H(\theta - \theta^*) + O(\|\theta - \theta^*\|^3). \tag{8}$$

Since $H$ is symmetric, it can be diagonalized as $H = Q\Lambda Q^\top$ with eigenvalues $\{\lambda_i\}$. In the eigenvector basis, the quadratic term becomes

$$(\theta - \theta^*)^\top H(\theta - \theta^*) = \sum_{i=1}^{d} \lambda_i(\theta_i - \theta_i^*)^2, \tag{9}$$

where $d$ denotes the dimension of the parameter vector $\theta$. Thus, near $\theta^*$, the loss can be written as

$$F(\theta) \approx F(\theta^*) + \frac{1}{2}\sum_{i=1}^{d} \lambda_i(\theta_i - \theta_i^*)^2 + O(\|\theta - \theta^*\|^3). \tag{10}$$

This expansion not only distinguishes between flat and sharp basins, depending on whether the dominant eigenvalue $\lambda_{\max}$ is small or large, but also establishes the foundation for later results where $\lambda_{\max}$ directly enters the expression for the sharp basins.

### A.1 ADAM OPTIMIZER DYNAMICS

Before analyzing the stochastic properties of Adam, we first recall its standard update dynamics. At each step $t$, Adam maintains exponential moving averages of the gradient (first moment) and the squared gradient (second moment). The parameter update is then obtained by scaling the momentum term by the inverse square root of the second moment:

$$\begin{aligned}
m_{t+1} &= \beta_1 m_t + (1 - \beta_1)g_t, \\
v_{t+1} &= \beta_2 v_t + (1 - \beta_2)g_t^2, \\
\theta_{t+1} &= \theta_t - \alpha \frac{m_{t+1}}{\sqrt{v_{t+1} + \epsilon}},
\end{aligned} \tag{11}$$

where $g_t = \nabla f_{B_t}(\theta_t)$ is the mini-batch gradient, $\beta_1, \beta_2 \in [0, 1)$ are the exponential decay coefficients (with typical values close to 1, e.g., $\beta_1 = 0.9$, $\beta_2 = 0.999$ in practice), $\alpha$ is the learning rate, and $\epsilon$ is a small constant for numerical stability.

In the following subsections, we examine the stochastic variance of the momentum (Appendix A.2) and the expectation of the second moment (Appendix A.3), which together provide the foundation for defining the effective temperature in Adam's dynamics.

### A.2 VARIANCE OF MOMENTUM TERM

Consider the momentum update in the Adam optimizer:

$$m_{i,t+1} = \beta_1 m_{i,t} + (1 - \beta_1)g_{i,t}, \tag{12}$$

where $m_{i,t}$ is the momentum term, $g_{i,t}$ is the mini-batch gradient, and $\beta_1 \in [0, 1)$ is the momentum coefficient. We define the deviation from the conditional expectation as

$$\delta m_{i,t} = m_{i,t} - \mathbb{E}[m_{i,t} \mid \{\theta_s\}_{s \leq t}], \tag{13}$$

which represents the stochastic fluctuation of the momentum. Assuming the mini-batch gradient can be decomposed as

$$g_{i,t} = \nabla F_i(\theta_t) + \xi_{i,t}, \quad \mathbb{E}[\xi_{i,t}] = 0, \quad \mathrm{Var}[\xi_{i,t}] = \frac{\sigma_i^2}{b}, \tag{14}$$

where $\xi_{i,t}$ denotes the gradient noise due to mini-batch sampling, whose variance scales inversely with the batch size $b$. In the quasi-static approximation, where momentum accumulation is negligible in long-term training dynamics, i.e., the long-term variance of momentum deviations is dominated by gradient noise, we have

$$m_{i,t} \approx g_{i,t} \quad \Longrightarrow \quad \delta m_{i,t} \approx g_{i,t} - \mathbb{E}[g_{i,t}] = \xi_{i,t}. \tag{15}$$

Therefore, the variance of the momentum deviation is directly given by the gradient noise:

$$\mathrm{Var}[\delta m_i] \approx \mathrm{Var}[\xi_i] = \frac{\sigma_i^2}{b}, \tag{16}$$

which indicates that, in the long-time limit, the stochastic fluctuations of momentum are entirely determined by the mini-batch gradient noise.

### A.3 SECOND MOMENT ANALYSIS

Consider the update of the second-moment term in Adam:

$$v_{i,t+1} = \beta_2 v_{i,t} + (1 - \beta_2) \left[\nabla F_i(\theta_t) + \xi_{i,t}\right]^2, \tag{17}$$

where $v_{i,t}$ is the second-moment accumulator, $\beta_2 \in [0, 1)$ is the exponential decay coefficient, and $\xi_{i,t}$ denotes the stochastic gradient noise with zero mean and variance $\sigma_i^2/b$.

To understand the contribution of the noise to the second moment, we expand the squared term:

$$\left[\nabla F_i + \xi_i\right]^2 = (\nabla F_i)^2 + 2\nabla F_i \xi_i + \xi_i^2. \tag{18}$$

Taking expectations eliminates the cross term due to $\mathbb{E}[\xi_i] = 0$, yielding

$$\mathbb{E}[v_{i,t+1}] = \beta_2 \, \mathbb{E}[v_{i,t}] + (1 - \beta_2) \left(\mathbb{E}[(\nabla F_i)^2] + \mathbb{E}[\xi_i^2]\right). \tag{19}$$

In the long-time limit, the expectation of the second-moment term satisfies

$$\langle v_i \rangle = \mathbb{E}[(\nabla F_i)^2] + \frac{\sigma_i^2}{b}, \tag{20}$$

which highlights that the second moment accumulates both the squared deterministic gradient and the variance of mini-batch noise.

### A.4 EFFECTIVE TEMPERATURE

We begin by isolating the stochastic part of Adam's update along coordinate $i$,

$$\Delta\theta_i^{\text{stoch}} = -\alpha \frac{\delta m_i}{\sqrt{v_i + \epsilon}}, \tag{21}$$

where $\delta m_i := m_i - \mathbb{E}[m_i \mid \{\theta_s\}_{s \leq t}]$ denotes the zero-mean fluctuation of the momentum term. Using the law of total variance and the fact that $\mathbb{E}[\delta m_i \mid v_i] = 0$, the variance of the stochastic increment can be written as

$$\text{Var}\left[\Delta\theta_i^{\text{stoch}}\right] = \alpha^2 \, \mathbb{E}\left[\frac{\text{Var}(\delta m_i \mid v_i)}{v_i + \epsilon}\right]. \tag{22}$$

Under an adiabatic (i.e., timescale separation) approximation, fluctuations of $v_i$ are small relative to its mean and we may replace $\text{Var}(\delta m_i \mid v_i)$ by $\text{Var}(\delta m_i)$ and $\mathbb{E}[1/(v_i + \epsilon)]$ by $1/(\langle v_i \rangle)$, yielding

$$\text{Var}\left[\Delta\theta_i^{\text{stoch}}\right] \approx \alpha^2 \frac{\text{Var}[\delta m_i]}{\langle v_i \rangle}. \tag{23}$$

To relate the discrete-time fluctuation in Eq. 23 to an effective diffusion in parameter space, we invoke the discrete Einstein relation. For a process with increments $\Delta X$ over time step $\Delta t$, the variance satisfies $\text{Var}[\Delta X] = 2D \, \Delta t$, where $D$ is the diffusion coefficient. In Adam, each update processes a mini-batch of size $b$, which means that the statistical information gathered in one step is equivalent to $b$ nominal steps of single-sample updates. From an information-theoretic perspective, the effective time step therefore scales inversely with $b$, so we set $\Delta t_{\text{eff}} = \Delta t_{\text{nominal}}/b$ with $\Delta t_{\text{nominal}} = 1$. Substituting this relation into the Einstein formula yields

$$D_i^{\text{eff}} = \frac{\text{Var}\left[\Delta\theta_i^{\text{stoch}}\right]}{2 \, \Delta t_{\text{eff}}} = \frac{b}{2} \text{Var}\left[\Delta\theta_i^{\text{stoch}}\right] \approx \frac{b \, \alpha^2}{2} \frac{\text{Var}[\delta m_i]}{\langle v_i \rangle}. \tag{24}$$

The expression in Eq. 24 provides the diffusion coefficient produced by Adam's stochastic updates. To normalize the influence of batching on exploration, the effective temperature is defined as

$$T_i^{\text{eff}} = D_i^{\text{eff}} / \sqrt{\text{Var}(\xi_i)} \approx \frac{b^{3/2}\alpha^2}{2\sigma_i} \frac{\text{Var}[\delta m_i]}{\langle v_i \rangle}. \tag{25}$$

This quantity measures the diffusion strength after correcting for the noise scale, providing a consistent metric for parameter-space exploration across different batch sizes.

Finally, under the long-time limit, the fluctuation of the momentum is dominated by mini-batch noise, i.e., $\text{Var}[\delta m_i] = \sigma_i^2/b$ (Eq. 16). Substituting this relation into Eq. 25 yields

$$T_i^{\text{eff}} \approx \frac{\alpha^2 \sigma_i \sqrt{b}}{2\langle v_i \rangle}. \tag{26}$$

Equation 26 provides a compact expression for the effective temperature that governs Adam's stochastic dynamics. Importantly, it depends both on the standard deviation of the gradient noise $\sigma_i$ and on the expected second-moment term $\langle v_i \rangle$, which encodes the local curvature of the loss landscape. This dependence forms the basis for distinguishing between flat and sharp basins in the Appendix B and for applying Kramers' escape theory in the Appendix C, ultimately leading to the calculation of the critical learning rate in the Appendix D.

## B  ANALYSIS OF LANDSCAPE BASINS

We now examine how the effective temperature behaves in different types of basins. The key difference lies in the structure of $\langle v_i \rangle$, which encodes the balance between curvature and stochastic noise as given in Eq. 20.

### B.1  FLAT BASINS

In flat basins of the loss landscape, gradients are negligible so that $\mathbb{E}[(\nabla F_i)^2] \approx 0$. According to Eq. 20, the expected second-moment term reduces to

$$\langle v_i \rangle \approx \frac{\sigma_i^2}{b}. \tag{27}$$

Substituting this into the expression for the effective temperature (Eq. 26) yields

$$T_i^{\text{eff,flat}} = \frac{\alpha^2 \sigma_i \sqrt{b}}{2 \cdot (\sigma_i^2/b)} = \frac{\alpha^2 b^{3/2}}{2\sigma_i}. \tag{28}$$

Thus, in flat basins, the effective temperature decreases with the noise amplitude $\sigma_i$ and grows as $\alpha^2 b^{3/2}$, reflecting a strong enhancement with batch size.

### B.2  SHARP BASINS

In contrast, near a sharp basin with dominant Hessian eigenvalue $\lambda_{\text{max}}$, the deterministic curvature contribution cannot be neglected. The relation between the squared gradient and the curvature can be directly understood from the quadratic expansion around a local basin (i.e., Eq. 8). In a single eigen-direction with eigenvalue $\lambda$, Eq. 10 reduces to

$$f(\theta) \approx \frac{1}{2}\lambda(\theta - \theta^*)^2. \tag{29}$$

The corresponding gradient and curvature are

$$\nabla f = \lambda(\theta - \theta^*), \qquad \nabla^2 f = \lambda. \tag{30}$$

Taking expectations gives

$$\mathbb{E}[(\nabla f)^2] = \lambda^2 \, \mathbb{E}[(\theta - \theta^*)^2]. \tag{31}$$

At a local minimum, which can be regarded as a steady state in equilibrium statistical physics, a quadratic potential $U(x) = \frac{1}{2}\lambda x^2$ yields a Boltzmann distribution with variance $\langle x^2 \rangle = T/\lambda$. By analogy, under the quasi-equilibrium approximation for Adam, the displacement variance satisfies

$$\mathbb{E}[(\theta - \theta^*)^2] = \frac{T_{\text{eff}}}{\lambda}. \tag{32}$$

Combining the two expressions yields the compact relation

$$\mathbb{E}[(\nabla f)^2] = T_{\text{eff}} \, \mathbb{E}[\nabla^2 f]. \tag{33}$$

In this regime, Eq. 20 gives

$$\langle v_i \rangle = T^{\text{eff}} \lambda_{\max} + \frac{\sigma_i^2}{b}. \tag{34}$$

For sharp basins, the leading-order relation underestimates the suppression by curvature. Following classic treatments of diffusion in steep potentials Kramers (1940); Hänggi et al. (1990); Risken (1989), we include the next-order correction, which introduces a quadratic self-consistency dependence in the denominator:

$$T^{\text{eff}} = \frac{C_{\text{neq}}}{\left(T^{\text{eff}} \lambda_{\max} + s\right)^2}, \tag{35}$$

with

$$C_{\text{neq}} = \frac{b^{1/2} \alpha^2 \sigma_i}{2}, \qquad s = \frac{\sigma_i^2}{b}. \tag{36}$$

When the curvature term dominates, i.e. $T^{\text{eff}} \lambda_{\max} \gg s$, the self-consistency reduces to

$$T^{\text{eff}} \approx \frac{C_{\text{neq}}}{(T^{\text{eff}} \lambda_{\max})^2}, \tag{37}$$

which implies the cubic equation

$$\lambda_{\max}^2 (T^{\text{eff}})^3 = C_{\text{neq}}. \tag{38}$$

Solving for $T^{\text{eff}}$ yields

$$T_i^{\text{eff,sharp}} = \left(\frac{b^{1/2} \alpha^2 \sigma_i}{2\lambda_{\max}^2}\right)^{1/3}. \tag{39}$$

In sharp basins, the effective temperature therefore grows only as the cube root of both $\alpha^2$ and $b^{1/2} \sigma_i$, while being strongly suppressed by the curvature $\lambda_{\max}$. This stark contrast with the flat case underpins the optimizer's preference for flat basins when learning rate $\alpha$ is tuned close to its critical value.

## C  KRAMERS ESCAPE THEORY

The effective temperature derived in the previous section directly determines the probability of escaping a potential well. According to the classical result of Kramers Kramers (1940), the escape rate from a barrier of height $\Delta E$ is

$$\Gamma = A \exp\left(-\frac{\Delta E}{T^{\text{eff}}}\right), \tag{40}$$

where $A$ is a prefactor that sets the attempt frequency. Substituting the expressions for $T^{\text{eff}}$ in the flat and sharp regimes yields two distinct scaling forms.

**Flat basins.** In flat basins, the effective temperature is $T_{\text{flat}} = \alpha^2 b^{3/2}/2$. The corresponding escape rate is therefore

$$\Gamma_{\text{flat}} = A \exp\left(-\frac{\Delta E}{T_{\text{flat}}}\right) = A \exp\left(-\frac{2\Delta E}{\alpha^2 b^{3/2}}\right). \tag{41}$$

For convenience we define the coefficient

$$C_{\text{flat}} = \frac{2\sigma \Delta E}{b^{3/2}}, \tag{42}$$

so that

$$\Gamma_{\text{flat}} = A \exp\left(\frac{-C_{\text{flat}}}{\alpha^2}\right). \tag{43}$$

**Sharp basins.** In sharp basins with dominant curvature $\lambda_{\max}$, the effective temperature is

$$T_{\text{sharp}} = \left( \frac{b^{1/2}\alpha^2\sigma}{2\lambda_{\max}^2} \right)^{1/3}.$$

Substituting into Kramers' formula gives

$$\Gamma_{\text{sharp}} = A \exp\left( -\frac{\Delta E}{T_{\text{sharp}}} \right) = A \exp\left( -\frac{\Delta E (2\lambda_{\max}^2)^{1/3}}{b^{1/6}\,\alpha^{2/3}\,\sigma^{1/3}} \right). \tag{44}$$

Defining

$$C_{\text{sharp}} = \Delta E \cdot \frac{(2\lambda_{\max}^2)^{1/3}}{b^{1/6}\,\sigma^{1/3}}, \tag{45}$$

this can be compactly expressed as

$$\Gamma_{\text{sharp}} = A \exp\left( -\frac{C_{\text{sharp}}}{\alpha^{2/3}} \right). \tag{46}$$

## D    CRITICAL LEARNING RATE

The preference of the optimizer for flat over sharp basins can be quantified by the ratio of escape rates. Using the results of the Appendix C, the relative escape probability is

$$R(\alpha) = \frac{\Gamma_{\text{flat}}}{\Gamma_{\text{sharp}}} = \exp\left( -\frac{C_{\text{flat}}}{\alpha^2} + \frac{C_{\text{sharp}}}{\alpha^{2/3}} \right). \tag{47}$$

The critical learning rate is identified with the value of $\alpha$ that maximizes $R(\alpha)$. Differentiating with respect to $\alpha$ gives

$$\frac{dR}{d\alpha} = R(\alpha) \left( \frac{2C_{\text{flat}}}{\alpha^3} - \frac{2C_{\text{sharp}}}{3\alpha^{5/3}} \right). \tag{48}$$

Setting this derivative to zero yields the balance condition

$$\frac{2C_{\text{flat}}}{\alpha_c^3} = \frac{2C_{\text{sharp}}}{3\alpha_c^{5/3}}, \tag{49}$$

so that

$$\alpha_c^{4/3} = \frac{3C_{\text{flat}}}{C_{\text{sharp}}}. \tag{50}$$

To evaluate the coefficient ratio, recall that

$$C_{\text{flat}} = \frac{2\sigma\Delta E}{b^{3/2}}, \qquad C_{\text{sharp}} = \Delta E \cdot \frac{(2\lambda_{\max}^2)^{1/3}}{b^{1/6}\sigma^{1/3}}.$$

Hence

$$\begin{aligned}
\frac{C_{\text{flat}}}{C_{\text{sharp}}} &= \frac{\frac{2\sigma\Delta E}{b^{3/2}}}{\Delta E \cdot \frac{(2\lambda_{\max}^2)^{1/3}}{b^{1/6}\sigma^{1/3}}} \\
&= \frac{2b^{1/6}\sigma^{4/3}}{b^{3/2}(2\lambda_{\max}^2)^{1/3}} \\
&= \frac{2\sigma^{4/3}}{b^{4/3}(2\lambda_{\max}^2)^{1/3}} \\
&= \frac{2\sigma^{4/3}}{b^{4/3}\,2^{1/3}\lambda_{\max}^{2/3}} \\
&= \frac{2^{2/3}\,\sigma^{4/3}}{b^{4/3}\,\lambda_{\max}^{2/3}}.
\end{aligned}$$

Substituting this into Eq. equation 50 gives

$$\alpha_c^{4/3} = \frac{3 \cdot 2^{2/3} \, \sigma^{4/3}}{b^{4/3} \, \lambda_{\max}^{2/3}}. \tag{51}$$

Taking the three-fourths power yields the closed form

$$\alpha_c = \left(3 \cdot 2^{2/3}\right)^{3/4} \frac{\sigma}{b \, \lambda_{\max}^{1/2}}, \tag{52}$$

so that the numerical result is

$$\alpha_c \approx 3.22 \, \frac{\sigma}{b \, \lambda_{\max}^{1/2}}. \tag{53}$$

**Verify it is a maximum.** To verify this extremum is a maximum, we consider the second derivative of $R(\alpha)$:

$$\frac{d^2 R}{d\alpha^2} = R(\alpha) \left(\phi''(\alpha) + (\phi'(\alpha))^2\right), \tag{54}$$

where $\phi(\alpha) = -\frac{C_{\text{flat}}}{\alpha^2} + \frac{C_{\text{sharp}}}{\alpha^{2/3}}$ is the exponent.

At the extremum, $\phi'(\alpha) = 0$, so the second derivative simplifies to

$$\frac{d^2 R}{d\alpha^2} = R(\alpha)\phi''(\alpha). \tag{55}$$

Since $R(\alpha) > 0$ for all learning rate $\alpha > 0$, the sign of $\frac{d^2 R}{d\alpha^2}$ is determined entirely by $\phi''(\alpha)$. The second derivative of the exponent is

$$\phi''(\alpha) = \frac{d^2}{d\alpha^2}\left(-\frac{C_{\text{flat}}}{\alpha^2} + \frac{C_{\text{sharp}}}{\alpha^{2/3}}\right) = -\frac{6C_{\text{flat}}}{\alpha^4} + \frac{10C_{\text{sharp}}}{9\alpha^{8/3}}. \tag{56}$$

Substituting $\alpha_c^{4/3} = 3C_{\text{flat}}/C_{\text{sharp}}$ yields

$$\phi''(\alpha_c) = -\frac{6C_{\text{flat}}}{\alpha_c^4} + \frac{10C_{\text{flat}}}{3\alpha_c^4} = -\frac{8C_{\text{flat}}}{3\alpha_c^4} < 0. \tag{57}$$

Since $C_{\text{flat}}$ and $\alpha_c$ are positive, this confirms that $R(\alpha)$ reaches a maximum at $\alpha_c$.

This derivation shows that $\alpha_c$ corresponds to the learning rate that maximizes the relative escape probability from sharp basins to flat basins, providing a principled alternative to heuristic schedules. We emphasize that the above escape-rate based argument assumes sufficiently batch sizes such that gradient noise is approximately Gaussian and time-scale separation for $(m, v)$ holds. In particular, the extreme cases, e.g., $b = 1$ and large batches (vanishing noise), lie outside this regime and the Kramers-type picture is no longer valid.

# E  MODEL DETAILS

In this section, we describe the model architectures and training setups used to empirically validate the theoretical predictions across both vision and language tasks.

| Model Name | Parameters | Input Size | Layers | Hidden Size |
|---|---|---|---|---|
| MLP-5layer | $\sim$47.4K | 784 (MNIST) | 5 (fc) | 50 |
| CNN-2conv | $\sim$2.12M | $32\times32\times3$ | 2 conv + 2 fc | 32, 64 (conv); 512 (fc) |
| WideResNet-28 | $\sim$36.48M | $32\times32\times3$ | 28 ($6n+4$, $n=4$) | 16, 160, 320, 640 |

Table 1: Model architectures and parameter counts for the MLP, CNN, and WideResNet models trained from scratch on MNIST and CIFAR-10 image classification tasks.

| Model Name | Parameters | Vocab Size | Layers | Hidden Size | Heads |
|---|---|---|---|---|---|
| BERT-base-uncased | $\sim$110M | 30,522 | 12 | 768 | 12 |
| GPT-2 (small) | $\sim$124M | 50,257 | 12 | 768 | 12 |
| TinyLlama-1B | $\sim$1.1B | 32,000 | 22 | 2048 | 32 |

Table 2: Pre-trained large language model architectures, vocabulary sizes, and parameter counts, employed in fine-tuning experiments for the SST-2 language classification task.

| $\alpha_c \times$ scaling factor ($\pm25\%$ error) | Adam | AdamW_warmup | AdamW_warmup + cosine_decay | AdamW_cosine_decay | Adafactor |
|---|---|---|---|---|---|
| $\times 10 \pm 25\%$ | $0.795 \pm 0.008$ | $0.776 \pm 0.009$ | $0.797 \pm 0.005$ | $0.810 \pm 0.006$ | $0.790 \pm 0.007$ |
| $\times 5 \pm 25\%$ | $0.827 \pm 0.006$ | $0.809 \pm 0.005$ | $0.828 \pm 0.006$ | $0.840 \pm 0.004$ | $0.822 \pm 0.008$ |
| $\times 2 \pm 25\%$ | $0.839 \pm 0.006$ | $0.823 \pm 0.007$ | $0.845 \pm 0.005$ | $0.851 \pm 0.005$ | $0.832 \pm 0.006$ |
| $\times 1.5 \pm 25\%$ | $0.867 \pm 0.009$ | $0.865 \pm 0.004$ | $0.891 \pm 0.005$ | $0.881 \pm 0.004$ | $0.867 \pm 0.007$ |
| $\times 1 \pm 25\%$ (*) | $\mathbf{0.869} \pm 0.006$ | $\mathbf{0.873} \pm 0.006$ | $\mathbf{0.899} \pm 0.004$ | $\mathbf{0.884} \pm 0.004$ | $\mathbf{0.870} \pm 0.004$ |
| $\times 0.5 \pm 25\%$ | $0.751 \pm 0.004$ | $0.729 \pm 0.009$ | $0.741 \pm 0.006$ | $0.753 \pm 0.003$ | $0.736 \pm 0.008$ |
| $\times 0.1 \pm 25\%$ | $0.664 \pm 0.008$ | $0.658 \pm 0.006$ | $0.661 \pm 0.005$ | $0.661 \pm 0.008$ | $0.654 \pm 0.004$ |

Table 3: Test accuracy (mean $\pm$ s.d.) for ResNet-28 trained on the CIFAR-10 dataset, evaluated across learning-rate scaling factors with a $\pm25\%$ estimation error over 10 independent runs.

# F    SUPPLEMENTARY FIGURES

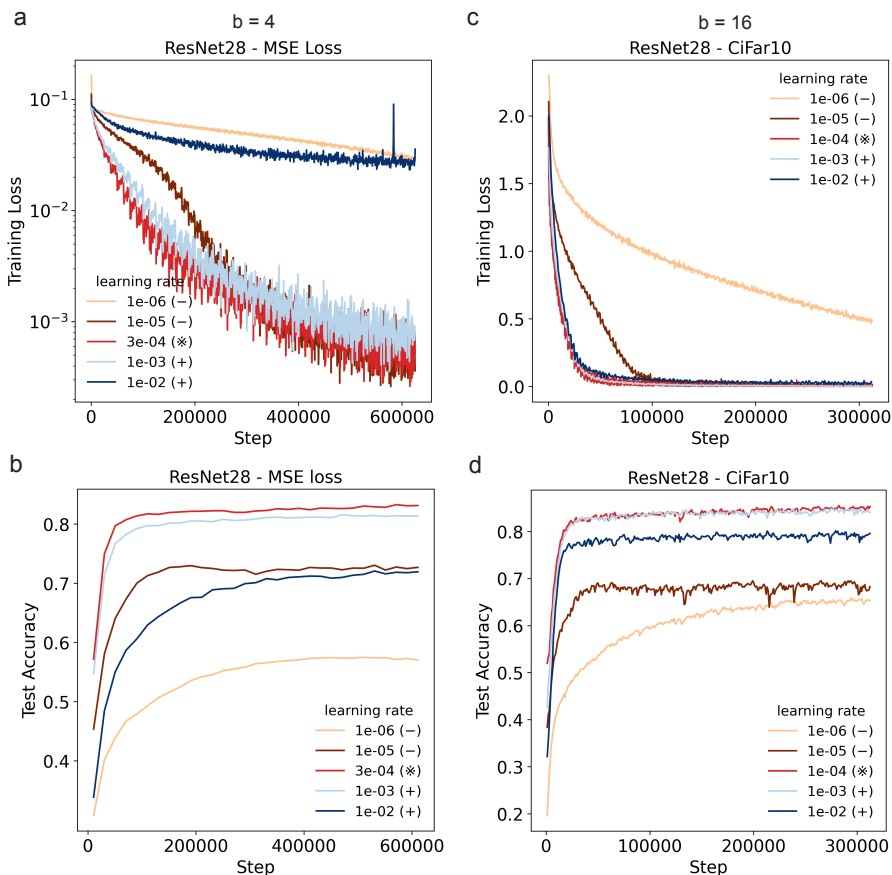

Figure 1: **Effect of batch size and loss function on the predictive critical learning rate.** Training loss (top) and test accuracy (bottom) for ResNet-28 on CIFAR-10 with different batch sizes. (**a**, **b**) Small batch size ($b = 4$) with Mean Squared Error (MSE) function. (**c**, **d**) Intermediate batch size ($b = 16$ with Cross-Entropy function). Curves correspond to learning rates below (–), above (+), and equal to (✳) the predicted critical value $\alpha_c$. In both settings, training with the theoretically predicted learning rate achieves better stability and test accuracy compared to substantially smaller or larger values.

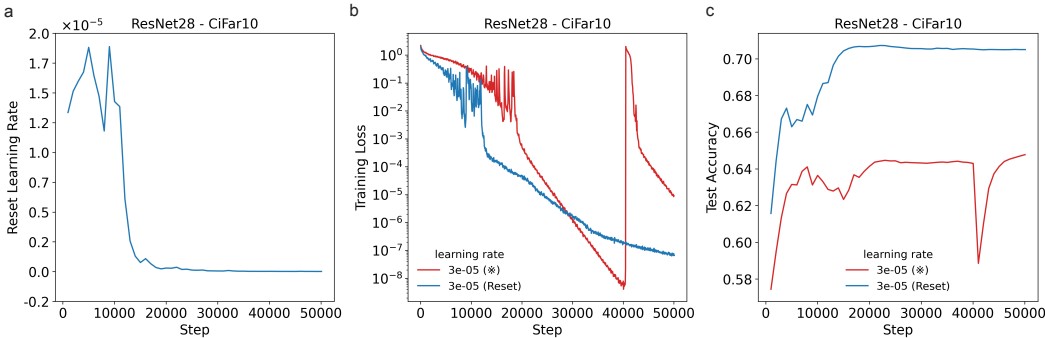

Figure 2: **Resetting the learning rate in large-batch training reveals sharp-basins confinement.**
Experiments with ResNet-28 on CIFAR-10 using batch size $b = 100$. (**a**) Locally re-estimated and
reset critical learning rate $\alpha_c$ during training, showing a sharp decay that indicates confinement in
narrow basins. (**b**) Training loss under fixed $\alpha_c$ (red, ✳) versus repeated resets to the locally re-
estimated $\alpha_c$ (blue). (**c**) Corresponding test accuracy. Unlike the small-batch case in Fig. 4(b,d,f),
where resets enabled escape and improved generalization, large-batch training locks the model into
sharp basins, leading to a lower and stagnant test performance. This contrast highlights how larger
batches reduce stochastic exploration and bias Adam toward sharp basins.

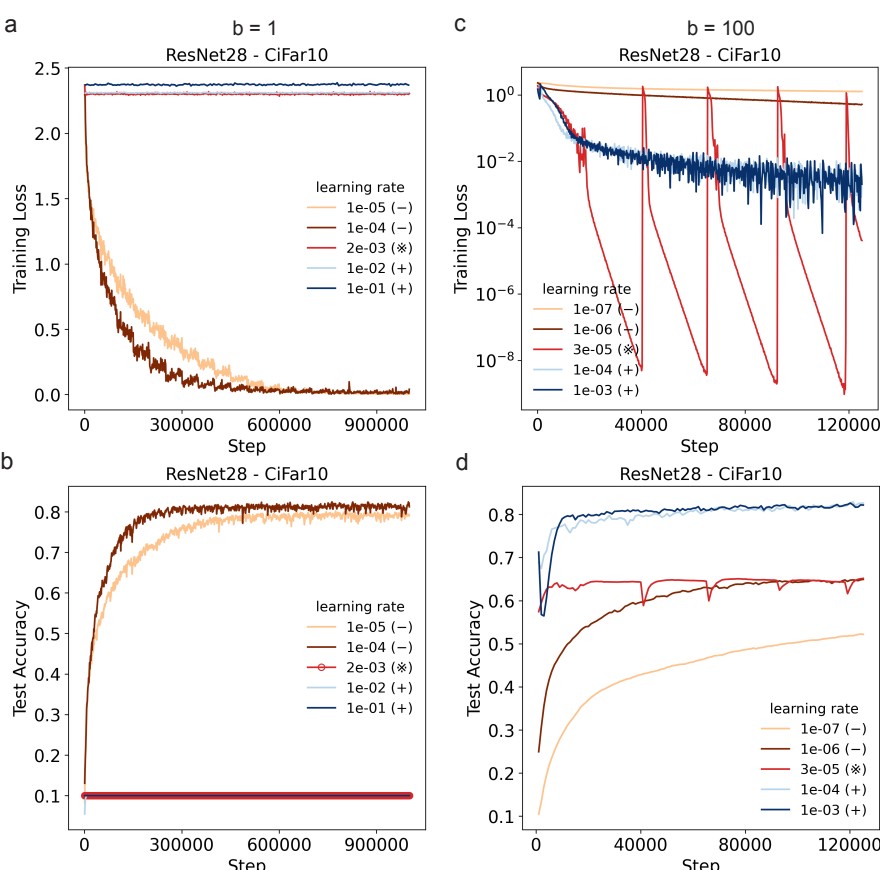

Figure 3: **Breakdown of the predictive critical learning rate at extreme batch sizes.** (**a**, **b**) ResNet-28 on CIFAR-10 with small batch size ($b = 1$). (**c**, **d**) ResNet-28 on CIFAR-10 with large batch size ($b = 100$). Training loss (top row) and test accuracy (bottom row) are plotted across different learning rates. The predicted $\alpha_c$ (❋) aligns poorly with optimal performance in both extremes: at $b = 1$, noise is non-Gaussian and convergence degrades, while $b = 100$, stochasticity vanishes, requiring larger-than-predicted learning rates to prevent instability.

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
