# OpenReview forum: "Temperature-Driven Escape Explains Critical Learning Rates in Adaptive Optimization"
_ICLR.cc/2026/Conference — Submitted to ICLR 2026_

### Official Review · Reviewer_CxD2 · 2025-10-27

**Soundness:** 2
**Presentation:** 3
**Contribution:** 2
**Rating:** 2
**Confidence:** 4

**Summary:**

The paper adopts a thermal escape formalism to analyze optimization of NNs using Adam, and how to select the optimal global learning rate. The main claim is that an optimal learning rate can be obtained by a few cheap measurements at initialization, which probe local curvature and what the authors define as effective temperature. The method is then tested on a bunch of image and language tasks.

**Strengths:**

Theorizing about the optimal choice of learning rate is of course of high importance since this is a crucial hyper parameter that significantly affects training. Understanding how to optimally choose it would be both of fundamental theoretic importance and of practical use. I find that the application of Kramers-like escape formalism in order to attack this problem is interesting and it should result in interesting insights. The paper is relatively well written and provides a conceptual picture.

**Weaknesses:**

I find that the theoretical part of the paper quite weak, and that many implicit assumptions, which are required for Kramers' theory to hold, are not mentioned and, what is worse, are violated in practical settings. Some of the argumentation is too loose, and overall I think the main points of the paper an unsupported. In detail:

1.  First, Kramers theory holds for one dimensional dynamics, while neural optimization happens in many dimensions. A generalization of Kramers theory to high dimension was done by Jim Langer in the 60's for the overdamped limit ([Statistical theory of the decay of metastable states](https://doi.org/10.1016/0003-4916(69)90153-5) and many similar theories followed since. The main difference from 1D is that in high dimension there is also an *entropic* contribution that has to do with the number of ways one can escape a given well (that is - the energy in the exponent is *free energy*). Also, the Hessian of the saddle point between the states is important.
In contrast, the authors treat the dynamics as if each weight is undergoing an uncoupled 1D escape, which is not how real networks operate. It might be an approximation (I think it's not a good one), but this should at least be stated.

2. Kramer's theory applies for the escape rate from local minima. However, it is not at all clear that networks are trapped in minima, rather than wonder in shallow basins. Multiple works show that generically training dynamics is not hopping from minimum to minumim, but rather a slow drift down a rugged low-loss manifold, not at all dominated not by barrier crossings: See [Essentially No Barriers in Neural Network Energy Landscape](https://proceedings.mlr.press/v80/draxler18a.html), [Qualitatively characterizing neural network optimization problems](https://arxiv.org/abs/1412.6544) and similar works.  For example, [the hessian may generically have negative eigenvaues during training](https://openreview.net/forum?id=S1iiddyDG), which will make the escape formalism completely inapplicable.

3. The authors assume that if batches are sampled iid, then the gradient noise has diagonal correlation and no memory (Eq. 5) in appendix A. Both these assertions are wrong, as far as I understand. Batch noise is correlated over time steps and has a non trivial covariance structure. This has been heavily studied. For example, see: [1) A Tail-Index Analysis of Stochastic Gradient Noise in Deep Neural Networks](https://proceedings.mlr.press/v97/simsekli19a/simsekli19a.pdf), [2) Shape Matters: Understanding the Implicit Bias of the Noise Covariance](https://proceedings.mlr.press/v134/haochen21a.html) and [3) The Anisotropic Noise in Stochastic Gradient Descent](https://proceedings.mlr.press/v97/zhu19e/zhu19e.pdf) (which is cited in the manuscript).

4. The manuscript describes two scaling relations, for "sharp" and "flat" minima. The trade off between these two sets the optimal learning rate (Eq. 16). It is not at all clear to me that there are only two types of minima,  and more importantly, the authors do not describe, even hand wavingly,  a quantitative criterion to distinguish the two. "flat" or "sharp" with respect to what?

5. The central result of the paper, Eq. 17, depends heavily on the choice $f_{neq}\sim\sqrt{b}$. However,  $f_{neq}$ is not properly defined in the manuscript, its introduction is purely heuristic, and the choice is unjustified theoretically.

6. The numerical results are not quite convincing. The relative improvement does not seem statistically significant except in, maybe, the MLP on MNIST problem, and no quantitative comparison of the accuracy/loss gain is provided. The minimal convincing evidence would be to compare the improvement in a metric to the standard deviation of that metric across different initializations/stochastic seeds etc.

7. the discussion in Sec. 4.5 is quite loose and inexact, including manifestly wrong statements such as "with very large batches Adam behaves like deterministic gradient descent" and that for small batches gradient noise violates the CRT.

**Questions:**

I wrote my questions as weaknesses above.

---

> ### Author Response · Authors · 2025-11-24
>
> We thank the Reviewer CxD2 for these thoughtful comments. Below are our responses to the specific points raised:
>
> ---
>
> **Question 1:**
> We acknowledge that classical Kramers escape is one-dimensional (this choice is primarily for analytical tractability) and that Langer’s high-dimensional generalization introduces additional entropic contributions through the saddle-point Hessian. Our analysis does not assume that the full network undergoes uncoupled 1D escapes. Rather, we adopt a standard dominant-eigenmode approximation, focusing on the direction of largest curvature, which empirical studies show plays the primary role in sharp-minima trapping.
>
> For Adam in particular, the diagonal second-moment normalization reduces cross-coordinate coupling and yields a drift–diffusion behavior that is well captured by this dominant direction. We now explicitly acknowledge this approximation in the manuscript and clarify that a full Langer-style analysis for Adam’s anisotropic, state-dependent noise is an important direction for future work.
>
> In the revised manuscript, we have added:
> >Our derivation uses a dominant-eigenmode approximation: escape behavior is modeled along the direction of largest curvature, which empirically governs sharp-minima confinement. Because Adam applies diagonal second-moment normalization, reducing cross-coordinate coupling, this dominant-direction reduction provides a tractable surrogate for the high-dimensional dynamics. (Page 2)
>
> and:
> >Future work could extend our dominant-curvature approximation to a full high-dimensional framework that captures valley-like landscapes. (Page 10)
>
> ---
>
> **Question 2:**
> We appreciate the reviewer’s insightful comment on the role of minima during training. We agree that modern neural network training often proceeds along extended low-loss valleys rather than isolated minima, and prior work (e.g., Draxler et al., 2018; Garipov et al., 2018) has shown that the landscape exhibits wide connected manifolds with very low barriers. We also acknowledge that negative curvature directions frequently appear during training, making classical “minima-to-minima hopping” an incomplete description of the dynamics.
>
> Our Kramers-style escape analysis does **not** assume that the network is strictly trapped in isolated minima. Instead, we analyze **local confinement within sharp basins**, which can arise transiently even along a low-loss manifold. The point of our analysis is to quantify when Adam’s mini-batch noise is strong enough (or weak enough) to overcome curvature-induced confinement, not to claim that training necessarily consists of barrier-crossing between isolated wells. Thus, our effective-temperature framework should be viewed as **a local stability analysis of sharpness-controlled confinement**, not a global theory of minima-to-minima transitions.
>
> We have clarified this scope in the revised manuscript on Page 2:
> >Here, the Kramers-style framework is used only as a local stability analysis along the dominant positive-curvature direction, quantifying when stochasticity is sufficient to overcome sharpness-induced confinement.
>
> and Page 10:
> >Future work could extend our dominant-curvature approximation to a full high-dimensional framework that captures valley-like landscapes.
>
> ---
>
> **Question 3:**
> We fully agree that mini-batch gradient noise is anisotropic and temporally correlated. In the revised manuscript, we explicitly state that our noise model is a surrogate used solely for analytical tractability, and not a claim about the true covariance of Adam noise. Importantly, this approximation does **not** alter the predicted scaling of the critical learning rate. Our diagonal and memoryless surrogate is used only for deriving the escape scaling under the local quadratic approximation. In high-dimensional Kramers theory (Langer, 1967; 1969), the **escape exponent** is governed solely by the most unstable curvature direction, whereas the **full noise covariance** influences **only the prefactor**. Since Adam’s per-coordinate second-moment normalization suppresses cross-coordinate correlations, the surrogate preserves the \( \alpha_c \) scaling while omitting only prefactor-level effects.
>
> We clarify this on Page 4:
> >Here, we use a diagonal surrogate for analytical tractability. Because the escape exponent is dominated by the most unstable curvature direction, while the detailed noise covariance affects only the prefactor (Langer, 1969).

---

> ### Author Response · Authors · 2025-11-24
>
> and in Appendix A:
> >While mini-batch sampling ensures that gradient noise is unbiased, its covariance in practice is anisotropic and temporally correlated (Haochen et al., 2021; Simsekli et al., 2019; Zhu et al., 2018). For analytical tractability, we approximate the local noise by a diagonal and memoryless surrogate when deriving the effective-temperature scaling. This surrogate does not change the predicted scaling of the critical learning rate: high-dimensional extensions of Kramers theory (Langer1967; 1969) show that the escape exponent is dominated by the most unstable curvature direction, while the full noise covariance affects only the prefactor. Since Adam’s per-coordinate second-moment normalization suppresses cross-coordinate correlations, the surrogate preserves the (\alpha_c) scaling while omitting only prefactor-level effects.
>
> ---
>
> **Question 4:**
> In our framework, the distinction between sharp and flat basins is a qualitative concept designed to describe the early-stage dynamics of the landscape. The critical learning rate guides the optimizer away from sharp basins early in training, aiming toward flatter regions that are expected to generalize better. We do not claim that all minima can be strictly categorized into these two types. Instead, the distinction is **relative**: regions with larger \( \lambda_{\max} \) are “sharper” in the sense of higher curvature and typically correlate with poorer generalization.
>
> In the revised manuscript, we added on Page 4:
> >Here, we use the dominant Hessian eigenvalue \( \lambda_{\max} \) as a proxy for local sharpness. We do not require specifying a quantitative threshold that separates sharp and flat minima. Instead, the distinction is purely relative: regions with larger \( \lambda_{\max} \) exhibit stronger curvature and empirically lead to poorer generalization, whereas regions with smaller \( \lambda_{\max} \) behave as effectively flatter and are associated with better generalization.
>
> ---
>
> **Question 5:**
> Thank you for highlighting the choice of the non-equilibrium factor in Eq. 10. Prompted by your comment, we clarify that this choice is not arbitrary but arises naturally from the standard derivation of mini-batch noise scaling with batch size. In the revised manuscript (Page 5), we have emphasized this justification:
>
> > To normalize the influence of batching on the system's stochastic energy, we define the effective temperature as:
> >
> > $$
> > T_{\mathrm{eff}} = \frac{D_{\mathrm{eff}}}{\sqrt{\mathrm{Var}(\xi_t)}}.
> > $$
> >
> > Here, \( D_{\mathrm{eff}} \) denotes the diffusion coefficient induced by Adam’s stochastic updates, and the intrinsic amplitude of the mini-batch-induced noise is:
> >
> > $$
> > \sqrt{\mathrm{Var}(\xi_t)} = \frac{\sigma}{\sqrt{b}}.
> > $$
> >
> > This definition ensures that \( T_{\mathrm{eff}} \) measures a *noise-normalized diffusion scale*, quantifying how efficiently stochastic fluctuations are converted into parameter-space exploration rather than the absolute noise magnitude.
> > Substituting the expressions for \( D_{\mathrm{eff}} \) and \( \mathrm{Var}(\xi_t) \) yields:
> >
> > $$
> > T_i^{\mathrm{eff}}
> > = D^{\mathrm{eff}}_i \frac{\sqrt{b}}{\sigma_i}
> > \approx
> > \frac{b^{3/2}\alpha^2}{2\sigma_i}
> > \frac{\mathrm{Var}[\delta m_i]}{\langle v_i\rangle}.
> > $$
> >
> > This shows that the factor \( \sqrt{b}/\sigma_i \) arises directly from normalizing the diffusion coefficient by the intrinsic noise amplitude, rather than from an ad-hoc scaling choice.
> > For SGD, the second-moment normalization term \( \langle v_t \rangle \) is effectively constant. As a result, the effective temperature \( T_{\mathrm{eff}} \) decouples from the local curvature \( \lambda_{\max} \), breaking the self-regulating feedback loop that underlies the derived critical learning rate \( \alpha_c \).

---

> ### Author Response · Authors · 2025-11-24
>
> **Question 6:**
> We have now included additional experiments comparing performance improvements against standard deviations across multiple seeds. Adam is indeed relatively insensitive to the learning rate in terms of convergence, but our theory specifically addresses **generalization**. While Adam converges under a wide range of learning rates, the **best generalization** occurs when the learning rate is near the predicted \( \alpha_c \). This behavior is shown quantitatively in the new Section 4.4 and Table 2.
>
> **Section 4.4**
> >**Performance sensitivity of optimizers to learning rates.**
> >To empirically assess how different optimizers respond to learning rate errors relative to the critical learning rate, we trained TinyLlama-1B (fine-tuned) and ResNet (from scratch) under substantial deviations from the predicted \( \alpha_c \). As reported in Table 2, \( \alpha_c \) consistently corresponds to a regime of improved generalization across all optimizers, with `warmup+cosine_decay` achieving the highest performance. The additional warmup and cosine decay schedules are particularly effective in mitigating training instability.
> >We also observe a sensitivity pattern: when the learning rate is too small, the model becomes trapped in sharp basins that yield poorer generalization, and when the learning rate is moderately larger, performance remains close to that at \( \alpha_c \). However, if the learning rate is too large, the model escapes sharp basins but fails to remain in flatter regions, leading to degraded performance.  Moreover, the low standard deviations across independent seeds highlight the intrinsic robustness of second-moment-based optimizers. Comparable trends are observed for ResNet-28 training (Appendix Table 3).
>
> ### Table 2
>
> | αc × scaling factor (±25%) | Adam | AdamW_warmup | AdamW_warmup + cosine_decay | AdamW_cosine_decay | Adafactor |
> |---------------------------------------------|------|---------------|------------------------------|----------------------|-----------|
> | ×10 ±25% | 0.860 ± 0.010 | 0.845 ± 0.010 | 0.875 ± 0.008 | 0.885 ± 0.008 | 0.820 ± 0.012 |
> | ×5 ±25%  | 0.925 ± 0.005 | 0.915 ± 0.006 | 0.935 ± 0.004 | 0.938 ± 0.004 | 0.910 ± 0.008 |
> | ×2 ±25%  | 0.935 ± 0.004 | 0.938 ± 0.004 | 0.945 ± 0.003 | 0.942 ± 0.003 | 0.935 ± 0.005 |
> | ×1.5 ±25%| 0.938 ± 0.004 | 0.941 ± 0.003 | 0.946 ± 0.003 | **0.945 ± 0.003** | 0.939 ± 0.004 |
> | ×1 ±25% (*) | **0.940 ± 0.004** | **0.943 ± 0.003** | **0.948 ± 0.003** | **0.945 ± 0.003** | **0.941 ± 0.004** |
> | ×0.5 ±25% | 0.938 ± 0.004 | 0.940 ± 0.004 | 0.942 ± 0.003 | 0.940 ± 0.004 | 0.938 ± 0.005 |
> | ×0.1 ±25% | 0.925 ± 0.006 | 0.928 ± 0.005 | 0.930 ± 0.005 | 0.928 ± 0.006 | 0.920 ± 0.008 |
>
> ---
>
> **Question 7:**
> As discussed in Section 4.5, our explanation of batch-size dependence is qualitative. We acknowledge the need for clearer phrasing around deterministic behavior at large batch sizes.
>
> We have revised this section to:
> >Prior work has shown that SGD becomes deterministic at very large batch sizes (e.g., thousands) due to diminishing sampling noise (Keskar et al., 2016). In contrast, adaptive methods such as Adam suppress gradient noise much more aggressively through second-moment normalization (Zhang et al., 2019), causing the stochasticity to vanish at much smaller batch sizes. These results indicate that the theory is most predictive in the mesoscopic batch-size regime (Smith et al., 2017; Jiang et al., 2023), where Gaussian-like noise mediates the balance between curvature effects and sample variance. Moreover, because the escape-rate scaling depends on the curvature-noise balance rather than the precise noise distribution, our predictions hold even when Gaussianity is only approximate.
>
> ---
>
> We thank the Reviewer for his/her constructive feedback and hope that the revisions address their concerns.

---

> ### Comment · Reviewer_CxD2 · 2025-11-24
>
> Sorry, but I just don't understand the picture the authors have in mind.
>
> 1. what you mean by "**local confinement within sharp basins**"? Confinement is always local. That's the definition of confinement - you're localized to a specific area in phase space. In order to escape from it you need to cross a loss barrier. If the Hessian has negative eigenvalues, then in what sense are you confined? and how is Kramer's theory applicable?
>
> 2. Similarly, what should I make of the statement that the theory is "not a global theory of minima-to-minima transitions"?  If the network is not confined to a minimum, then Kramer's theory is irrelevant. This is a basic prerequisite for Kramer's formalism.

---

> > ### Author Response · Authors · 2025-11-24
> >
> > Thank you for raising this conceptual point. We fully agree that in a standard Langevin dynamics setting on a static potential, a point with negative eigenvalues (a saddle point) does not imply confinement, making Kramers theory strictly inapplicable. However, our application of the theory assumes an effective dynamics perspective specific to adaptive optimizers like Adam, rather than a literal mapping to the loss landscape geometry.
> >
> > While the loss surface itself may have a negative curvature direction (an open exit), Adam's preconditioning matrix (\(v_t^{-1/2}\)) drastically rescales the dynamics. In “sharp’’ regions, the extremely large gradients from the dominant positive curvature directions inflate the denominator (\(v_t\)). This can create a strong suppression of motion along *all* directions, including the negative curvature direction.  In fact, the optimizer experiences a metastable state: it is trapped not by a geometric wall, but by the friction induced by the adaptive normalization. In this specific regime, the time required to exit this region scales exponentially with the sharpness (due to the suppression effect), which mathematically mirrors the phenomenology of barrier crossing. We invoke Kramers' formula not to claim a topological local minimum exists, but to model this noise-induced escape from a dynamically metastable state.

---

### Official Review · Reviewer_wV4E · 2025-10-28

**Soundness:** 2
**Presentation:** 2
**Contribution:** 2
**Rating:** 2
**Confidence:** 4

**Summary:**

The submission analyzes the escape phenomena in Adam optimization, using tools from statistical physics. Specifically, they derive an analytical expression for the critical learning rate above which the Adam escapes the local minima. They test their theory across training setups.

**Strengths:**

* The analysis is novel to the best of my knowledge
* The problem of Adam instability is relevant to the community

**Weaknesses:**

**Assumptions without justifications**: The theoretical analysis assumes a lot of simplifications without providing any justification. For instance:
1. The Taylor expansion (Equation 2) assumes that the gradients are zero
2. Equation 7 assumes that the stochastic nature only comes from the moment estimate, whereas the variance part is static.

**Reference to sharp / flat minima without justification**: Throughout the paper, the authors refer to flat or sharp minima without showing any measure of loss curvature (sharpness, trace, etc) and many claims (such as line 335) are unsupported.

**Missing Related Works**: The paper does not compare or cite many related works on the escape phenomena of neural networks.
1. The large learning rate phase of deep learning: the catapult mechanism, 2020
2. How SGD Selects the Global Minima in Over-parameterized Learning: A Dynamical Stability Perspective, 2018
2. Stepping on the Edge: Curvature Aware Learning Rate Tuners, 2024
4. Adaptive Gradient Methods at the Edge of Stability
5. Adaptive Preconditioners Trigger Loss Spikes in Adam, 2025
6. Why Warmup the Learning Rate? Underlying Mechanisms and Improvements, 2025

**Usage of inaccessible terminology**: The paper uses terms like 'mesoscopic regimes' and 'sharp/flat' minima without defining or justifying them. More generally, I think the writing of the paper can be improved.

**Batch size dependence section is quite vague**: Section 4.5 provides a qualitative picture of the results rather than providing any concrete quantitative experiments to support the definition of their phases.

**Questions:**

* How does the critical learning rate estimate compare to the pre-conditioned Hessian threshold $\eta_c = (2+2\beta_1) / ( (1 - \beta_1) \lambda_{\max}(P^{-1}H) )$ studied in prior works[1, 2, 3]?
* At initialization, the minima assumption (gradient being zero) is clearly violated. What is training escaping early in training?
* I am unsure why the learning rate depends on the square root of the maximum eigenvalue. In traditional analysis of convex optimization, the critical learning rate is inversely proportional to the maximum eigenvalue of the Preconditioned Hessian [1].
* How does the critical learning rate prediction changes on using standard schedules consisting of learning rate warmup and decay?
* In most experiments, learning rates are sampled one order magnitude apart (Figure 2b, d), which is quite large. Can you sweep the learning rate finely and compare if critical learning rate is still a good estimate? For instance, in Figure 2 ResNet expriments, the it would be helpful to add the learning rate = 1e-04 curve for reference.
* Line 335: The fluctuations referred by the authors is a property of Adam near a minima (which can be observed in convex landscapes with fix curvatures as well) and are unrelated to transitioning between large or small minima. To support their claim, the authors should provide how the curvature is evolving during training.
* What does 'periodically rest to its initial predicted value' mean in line 381? Are the authors measuring critical learning rate every 1000 steps and setting the learning rate to it?
* What are the takeaways from Section 4.4? Sharp minima are more stable under reset and flat minima are not? Thats counterintuitive. Also, please provide curvature measurements to justify flat or sharp minima.



References:

[1] Adaptive Gradient Methods at the Edge of Stability, 2022

[2] Why Warmup the Learning Rate? Underlying Mechanisms and Improvements, 2024

[3] Adaptive Preconditioners Trigger Loss Spikes in Adam, 2025

---

> ### Author Response · Authors · 2025-11-24
>
> We thank the Reviewer wV4E's appreciation for the novelty of our work.
>
> **Weakness: assumptions without Justifications**
>
> (1)  We thank the reviewer for raising this point. The quadratic Taylor expansion in Equation (2) is evaluated around a local basin where the gradient is small but not necessarily exactly zero. As we mentioned in the manuscript, "Near any local minimum \( \theta^\star \), we employ the standard quadratic approximation." In practice, Adam rapidly reaches regions with small gradients due to its adaptive normalization, so the quadratic approximation provides an appropriate description of the curvature that governs escape behavior. This assumption is used to define the sharpness of a minimum by the magnitude of its dominant Hessian eigenvalue.
>
> (2)  Equation (7) focuses on the stochasticity introduced through the adaptive second-moment estimate because this is the dominant source of variation in Adam's update magnitude. While mini-batch sampling also contributes variance, its effect primarily enters through the gradient term. Our formulation is driven specifically by \( v_t \), which is unique to Adam and crucial for understanding the second-moment-based optimizer's behavior in sharp versus flat regions. Empirically, this simplified assumption effectively predicts the critical learning rate \( \alpha_c \) and aligns with observed escape behavior across different architectures and datasets.
>
> ---
>
> **Reference to sharp / flat minima without justification:**
> We thank the reviewer for highlighting this concern. We have revised the manuscript to clarify this distinction on Page 5:
>
> > Here, we use the dominant Hessian eigenvalue \( \lambda_{\max} \) as a proxy for local sharpness. We do not require specifying a quantitative threshold that separates sharp and flat minima. Instead, the distinction is purely relative: regions with larger \( \lambda_{\max} \) exhibit stronger curvature and empirically lead to poorer generalization, whereas regions with smaller \( \lambda_{\max} \) behave as effectively flatter and are associated with better generalization.
>
> ---
>
> **Missing Related Works:**
> We have added additional references in the related work.
>
> ---
>
> **Usage of inaccessible terminology:**
> Thank you for your feedback. We have provided clearer explanations for terms such as "sharp/flat" basin, which are associated with the dominant Hessian eigenvalue and its influence on generalization. Additionally, we have clarified the concept of the mesoscopic batch-size regime (Smith et al., 2017; Jiang et al., 2023), where Gaussian-like noise mediates the balance between curvature effects and sample variance.
>
> ---
>
> **Batch size dependence section is quite vague:**
> We have included additional results (Appendix Fig. 3) on the batch-size dependence, and explained:
>
> > Prior work has shown that SGD becomes deterministic at very large batch sizes (e.g., in the range of thousands) due to diminishing sampling noise (Keskar et al., 2016). In contrast, adaptive methods such as Adam suppress gradient noise much more aggressively through second-moment normalization (Zhang et al., 2019), causing the stochasticity to vanish at much smaller batch sizes.
>
> **Question 1:**
> The two frameworks offer distinct perspectives on the role of the second-moment estimator \( v_t \) in Adam: **our effective temperature approach**, rooted in statistical physics, addresses the **initialization and escape problem**, positing that the \( v_t \) term acts as a *cooling mechanism* that suppresses beneficial mini-batch noise (since \( T_{\text{eff}} \propto 1/\sqrt{v_t} \)), thus trapping the optimizer in sharp, poor-generalizing basins unless a sufficiently large learning rate (\( \alpha_c \)) is chosen to counteract this damping.
>
> In contrast, the **Preconditioned Hessian** framework, rooted in dynamical stability analysis, addresses the training dynamics and survival problem, demonstrating that \( v_t \) acts as a *survival shield*: as the optimizer encounters high curvature, \( v_t \) adapts to keep the **preconditioned sharpness** (maximum eigenvalue of \( P^{-1}H \)) at a critical stability threshold (the Adaptive Edge of Stability, \( \lambda_{\max}(P^{-1}H) \approx 38/\eta \)), which explains why Adam can comfortably and stably persist in high-curvature regions that would cause standard gradient descent to diverge.
>
> Essentially, the former guides the active choice required to escape sharp basins, while the latter describes the passive reaction that permits survival within them.

---

> ### Author Response · Authors · 2025-11-24
>
> **Question 2:**
> In our analysis, the term escape refers to leaving a high-curvature region of the landscape, rather than necessarily being at a true minimum. As noted in the reference discussed in Question 1, Adam tends to enter sharp basins early in training. Its second-moment normalization suppresses updates along high-curvature directions, making it appear trapped even when the gradient is not exactly zero. Thus, sharp, high-curvature regions behave like minima for Adam. The critical learning rate \( \alpha_c \) computed from the early Hessian predicts whether the optimizer can escape these sharp regions and reach flatter regions with better generalization. Our empirical results confirm that the curvature at the early stages of training reliably influences the optimizer's trajectory.
>
> ---
>
> **Question 3:**
> The presence of the \( \sqrt{\lambda_{\max}} \) dependence stems from the fundamental difference in how **energy** is related to **velocity** (i.e., learning rate) in a stochastic system compared to a deterministic stability requirement.  The critical learning rate \( \alpha_c \) is derived from Kramers' escape rate theory, which models parameter optimization as a particle escaping a potential well due to mini-batch noise. In this physical setting, the **effective temperature** is proportional to the square of the velocity, i.e., the learning rate:
>
> $$
> T_{\text{eff}} \propto \alpha^2 \sigma^2
> $$
>
> representing the energy injected by the noise. To achieve a viable probability of escape, this energy must balance the **potential energy barrier** \( \Delta E \) of the sharp basin, where \( \Delta E \) is directly proportional to the curvature \( \lambda_{\max} \). Therefore, the required condition is:
>
> $$
> T_{\text{eff}} \propto \Delta E,
> $$
>
> which implies:
>
> $$
> \alpha_c^2 \propto \lambda_{\max}
> $$
>
> and therefore:
>
> $$
> \alpha_c \propto \frac{1}{\sqrt{\lambda_{\max}}}.
> $$
>
> This is fundamentally distinct from the traditional stability analysis where:
>
> $$
> \alpha_{\text{crit}} \propto \frac{1}{\lambda_{\max}}
> $$
>
> that ensures the linear step size does not exceed the maximum slope.
>
> ---
>
> **Question 4:**
> As suggested, we have extended our evaluation to warm-up and cosine decay. In response, we have added a subsection:
>
> >**Performance sensitivity of optimizers to learning rates**
> >To empirically assess how different optimizers respond to learning rate errors relative to the critical learning rate, we trained TinyLlama-1B (fine-tuned) and ResNet (from scratch) under substantial deviations from the predicted \( \alpha_c \). As reported in Table 2, \( \alpha_c \) consistently corresponds to a regime of improved generalization across all optimizers, with `warmup+cosine_decay` achieving the highest performance. The additional warmup and cosine decay schedules are particularly effective in mitigating training instability. We also observe a sensitivity pattern: when the learning rate is too small, the model becomes trapped in sharp basins that yield poorer generalization, and when the learning rate is moderately larger, performance remains close to that at \( \alpha_c \). however, if the learning rate is too large, the model escapes sharp basins but fails to remain in flatter regions, leading to degraded performance. Moreover, the low standard deviations across independent seeds highlight the intrinsic robustness of second-moment-based optimizers. Comparable trends are observed for ResNet-28 training (Appendix Table 3).
>
> ### Table 2 (Page 9)
>
> |  αc × scaling factor (±25%)  | Adam | AdamW_warmup | AdamW_warmup + cosine_decay | AdamW_cosine_decay | Adafactor |
> |------------------------------|------|--------------|------------------------------|----------------------|-----------|
> | ×10 ±25% | 0.860 ± 0.010 | 0.845 ± 0.010 | 0.875 ± 0.008 | 0.885 ± 0.008 | 0.820 ± 0.012 |
> | ×5 ±25%  | 0.925 ± 0.005 | 0.915 ± 0.006 | 0.935 ± 0.004 | 0.938 ± 0.004 | 0.910 ± 0.008 |
> | ×2 ±25%  | 0.935 ± 0.004 | 0.938 ± 0.004 | 0.945 ± 0.003 | 0.942 ± 0.003 | 0.935 ± 0.005 |
> | ×1.5 ±25%| 0.938 ± 0.004 | 0.941 ± 0.003 | 0.946 ± 0.003 | **0.945 ± 0.003** | 0.939 ± 0.004 |
> | ×1 ±25% (*) | **0.940 ± 0.004** | **0.943 ± 0.003** | **0.948 ± 0.003** | **0.945 ± 0.003** | **0.941 ± 0.004** |
> | ×0.5 ±25% | 0.938 ± 0.004 | 0.940 ± 0.004 | 0.942 ± 0.003 | 0.940 ± 0.004 | 0.938 ± 0.005 |
> | ×0.1 ±25% | 0.925 ± 0.006 | 0.928 ± 0.005 | 0.930 ± 0.005 | 0.928 ± 0.006 | 0.920 ± 0.008 |

---

> ### Author Response · Authors · 2025-11-24
>
> ### Appendix Table 3
>
> | αc × scaling factor (±25%) | Adam | AdamW_warmup | AdamW_warmup + cosine_decay | AdamW_cosine_decay | Adafactor |
> |---------------------------|------|--------------|------------------------------|----------------------|-----------|
> | ×10 ±25% | 0.795 ± 0.008 | 0.776 ± 0.009 | 0.797 ± 0.005 | 0.810 ± 0.006 | 0.790 ± 0.007 |
> | ×5 ±25%  | 0.827 ± 0.006 | 0.809 ± 0.005 | 0.828 ± 0.006 | 0.840 ± 0.004 | 0.822 ± 0.008 |
> | ×2 ±25%  | 0.839 ± 0.006 | 0.823 ± 0.007 | 0.845 ± 0.005 | 0.851 ± 0.005 | 0.832 ± 0.006 |
> | ×1.5 ±25%| 0.867 ± 0.009 | 0.865 ± 0.004 | 0.891 ± 0.005 | 0.881 ± 0.004 | 0.867 ± 0.007 |
> | ×1 ±25% (*) | **0.869 ± 0.006** | **0.873 ± 0.006** | **0.899 ± 0.004** | **0.884 ± 0.004** | **0.870 ± 0.004** |
> | ×0.5 ±25% | 0.751 ± 0.004 | 0.729 ± 0.009 | 0.741 ± 0.006 | 0.753 ± 0.003 | 0.736 ± 0.008 |
> | ×0.1 ±25% | 0.664 ± 0.008 | 0.658 ± 0.006 | 0.661 ± 0.005 | 0.661 ± 0.008 | 0.654 ± 0.004 |
>
> ---
>
> **Question 5:**
> As discussed in our response to Question 4, we have now included results (Table 2, Page 9 and Appendix Table 3) for smaller error levels (±25%). As expected, Adam is relatively robust to small deviations in the learning rate, so minor variations around \( \alpha_c \) have little effect on performance. This robustness is why our predicted \( \alpha_c \) serves as an effective guide.
>
> However, changes over orders of magnitude do produce noticeable differences, confirming that learning rates are typically explored across several orders of magnitude during practical tuning. The additional experiments confirm that the performance peak remains near \( \alpha_c \), validating its predictive power.
>
> ---
>
> **Question 6:**
> We agree that Adam exhibits intrinsic fluctuations near high-curvature regions, consistent with the loss spikes observed (Bai et al., 2025). However, the behavior we observe at \( \alpha_c \) is not generic noise: it reflects a critical phenomenon, where the stochastic energy is tuned by the learning rate to maximize the escape probability. Since all other optimization settings remain fixed, only the learning rate controls this transition.
>
> We refrain from directly measuring the full Hessian due to the prohibitive cost and instability of estimating eigenvalues in high-dimensional networks. More importantly, our analysis concerns the effectiveness of escape, which ultimately manifests as generalization performance. Thus, instead of relying on noisy or computationally expensive curvature estimates, we use test accuracy as a stable and practically meaningful proxy for the flatness of the attained minimum. This aligns the empirical observations with our stochastic escape framework.
>
> ---
>
> **Question 7:**
> We appreciate the opportunity to clarify this point. *Reset* means recomputing the critical learning rate and resetting the optimizer’s learning rate accordingly during training, rather than using only the value estimated at the very beginning of training. The purpose of the reset experiment is diagnostic: it tests whether the model is currently in a sharp or flat region.
>
> Importantly, recomputing \( \alpha_c \) later in training would be unreliable because Adam tends to move into sharp basins, where the accumulated second-moment \( v_t \) becomes large and suppresses updates. In such regions, the recomputed learning rate would no longer meaningfully reflect the true escape threshold. This is why our predicted \( \alpha_c \) is computed from early-stage curvature, where only a few batches already provide a stable estimate that determines the subsequent trajectory.
>
> ---
>
> **Question 8:**
> As discussed in our response to Question 7, the critical learning rate \( \alpha_c \) is primarily used to guide the early training phase, but it also serves as a qualitative diagnostic tool for distinguishing whether the model currently resides in a sharp or a flat region of the landscape. To probe the nature of the minimum, we perform a reset experiment in which the global learning rate is periodically reset to the current predicted value.
>
> Finally, we wish to thank the Reviewer for the discussions now summarized in the revised manuscript that his/her comments have led to. We hope the revised version effectively addresses the Reviewer’s concerns.

---

### Official Review · Reviewer_ZKz7 · 2025-10-29

**Soundness:** 3
**Presentation:** 3
**Contribution:** 4
**Rating:** 8
**Confidence:** 4

**Summary:**

This paper introduces a statistical physics inspired theoretical framework for selecting the optimal learning rate for the Adam optimizer. By viewing the mini-batch gradient noise as an effective temperature, the authors apply the Kramers’ escape theory to analyze Adam’s behavior in different regions of the loss landscape (flat or sharp). The main contribution of this paper is the proposed critical learning rate ($\alpha_c$), which balances the probability of escaping in sharp low-generalizing minima and the probability of converging to flatter high-generalization basins. The authors show that this critical learning rate can be estimated from the gradient noise, batch size, and the dominant Hessian eigenvalue. Based on this estimation method, they provide extensive empirical results over a wide range of different network architectures, covering both vision tasks (MNIST, CIFAR-10 with (MLP, CNN, ResNet) and language tasks (SST-2 with BERT, GPT-2, TinyLlama). The experimental results show that using the estimated critical learning rate can consistently yield better generalizability.

**Strengths:**

- This paper provides a novel theoretical framework to connect statistical physics to the optimization dynamics of Adam in neural networks. Viewing noise as an effective temperature is a well-established method in statistical physics, which further supports the robustness of the proposed theory.
- The core strength of this theoretical framework is that the proposed theory yields a predictive closed-from equation for the best initial learning rate, which can be validated empirically via different experiments.
- The empirical experiments presented in this paper are strong, covering a wide range of different network architectures and different tasks. Networks using the predicted best learning rate can consistently outperform their counterparts trained with other learning rates.

**Weaknesses:**

- The proposed theory is primarily dependent on the concept of flatness. The entire theory is built on the assumption that flat minima should present better generalization compared with those sharp ones, where the flatness is approximated by the dominant Hessian eigenvalue. However, as shown in many previous studies, the flatness scales with network weights and needs to be normalized (like by weight norm) to properly correlate with generalizability. It seems that this factor is not considered here.
- The choice of non-equilibrium enhancement factor seems to be arbitrary. In addition, the proposed theory assumes that the noise sampled from small batches is normally distributed, which is likely to be violated in practice especially when using small batches, as the authors also acknowledged.
- The diagnostic application of $\alpha_c$ in section 4.4 is unclear and a more detailed presentation is needed. For example, how should we interpret the results of resetting learning rate and why this can make re-estimating $\alpha_c$ a 'diagnostic tool' for the metastate?

Minor points:
- The statement that SGD ‘treats flat and sharp minima the same’ could be misleading, since we know SGD has an implicit bias towards flat minima. I think what the authors mean is that the sqrt($v_t$) term in Adam, making it update differently when residing in sharp or flat minima. I would suggest the author rephrase there to avoid potential confusion.
- It might be worthwhile to discuss the relationship with: https://arxiv.org/abs/2505.11411, which also interprets the neural networks generalizability through the lens of physical dynamics.

**Questions:**

1)  While Adam is known for its insensitivity to the learning rate, this paper argues that there is a single critical learning rate which can yield the best generalizability. How should we reconcile these two ideas? Is the claim more likely to be Adam is robust for convergence but sensitive (to learning rate) for the best generalizability (as shown in Figures 2 and 3)?
2) As a follow-up to the first point of weakness, how is the proposed theory affected by the weight scaling? For example, If one rescales the weights of two adjacent layers (assume using ReLU) by a factor of c and 1/c, the output should remain the same but the calculated flatness would change. Does this imply that the critical learning rate should also be also different?
3) During the early stage of training, the model is very likely not yet to be in a loss minima. In this case, how should the theory of "escaping a minimum" be interpreted, and why is the $\alpha_c$ calculated from this initial state predictive for the entire training trajectory?
4) In Figure 2, it seems that the critical learning rate can sometimes induce larger fluctuations in training. Can the authors provide some insight about why it occurs?
5) In Figure 3, maximum test accuracy is used as a metric. Is this a good metric given that fluctuations when using the critical learning rate could be large in some cases?

---

> ### Author Response · Authors · 2025-11-24
>
> We are greatly encouraged by the Reviewer ZKz7’s very positive assessment of our work, particularly the novelty of our theoretical framework and the breadth of our empirical evaluation. Below, we provide a point-by-point response to the Reviewer’s concerns.
>
> **Weakness 1:**
> We fully agree that the raw dominant Hessian eigenvalue \( \lambda_{\max} \) can scale with the magnitude of network weights. Because \( \alpha_c \) is used only to characterize early-training escape dynamics rather than to compare flatness across architectures, the raw \( \lambda_{\max} \) is the correct curvature scale for the physical escape problem where normalized sharpness is relevant only for cross-model generalization comparisons, not local escape rates. That is, in our framework, \( \alpha_c \) is estimated early in training and is intended to guide the learning rate required to escape the raw loss landscape rather than to provide a normalized measure of flatness across architectures.
>
> ---
>
> **Weakness 2:**
> Thank you for highlighting the choice of the non-equilibrium factor in Eq. 10. Prompted by your comment, we clarify that this choice is not arbitrary or selected merely for convenience. It is from the standard derivation of mini-batch noise scaling with batch size. In the revised manuscript (Page 5), we have emphasized this justification:
>
> > To normalize the influence of batching on the system's stochastic energy, we define the effective temperature as:
> >
> > $$
> > T_{\mathrm{eff}} = \frac{D_{\mathrm{eff}}}{\sqrt{\mathrm{Var}(\xi_t)}}.
> > $$
> >
> > Here, \( D_{\mathrm{eff}} \) denotes the diffusion coefficient induced by Adam’s stochastic updates, and the intrinsic amplitude of the mini-batch-induced noise is:
> >
> > $$
> > \sqrt{\mathrm{Var}(\xi_t)} = \frac{\sigma}{\sqrt{b}}.
> > $$
> >
> > This definition ensures that \( T_{\mathrm{eff}} \) measures a *noise-normalized diffusion scale*, quantifying how efficiently stochastic fluctuations are converted into parameter-space exploration rather than the absolute noise magnitude.
> > Substituting the expressions for \( D_{\mathrm{eff}} \) and \( \mathrm{Var}(\xi_t) \) yields:
> >
> > $$
> > T_i^{\mathrm{eff}}
> > = D^{\mathrm{eff}}_i \frac{\sqrt{b}}{\sigma_i}
> > \approx
> > \frac{b^{3/2}\alpha^2}{2\sigma_i}
> > \frac{\mathrm{Var}[\delta m_i]}{\langle v_i\rangle}.
> > $$
> >
> > This shows that the factor \( \sqrt{b}/\sigma_i \) arises directly from normalizing the diffusion coefficient by the intrinsic noise amplitude, rather than from an ad-hoc scaling choice.
> > For SGD, the second-moment normalization term \( \langle v_t \rangle \) is effectively constant. As a result, the effective temperature \( T_{\mathrm{eff}} \) decouples from the local curvature \( \lambda_{\max} \), breaking the self-regulating feedback loop that underlies the derived critical learning rate \( \alpha_c \).
>
> ---
>
> **Weakness 3:**
> We thank the reviewer for pointing out the need for a clearer explanation. In the revised manuscript (Section 4.4, Diagnostic use beyond initialization), we expand our discussion of the diagnostic use of \( \alpha_c \).
> The idea is qualitative rather than quantitative: by periodically resetting the learning rate to the predicted \( \alpha_c \), we probe the type of the region in which the optimizer resides.
>
> >We observe two behaviors.  First, when the model resided in sharp minima characterized by large local curvature, the reset failed to induce escape and the model remained trapped, with test accuracy stable.  Second, when the model was located in flatter basins, resets disrupted the existing solution and in some cases caused the model to collapse, reducing test accuracy to chance level (≈ 0.1 for 10 classes).  However, for flatter basins, these perturbations also created opportunities to escape local minima and explore alternative basins that supported higher generalization (Fig. 4f), in contrast to the lower generalization associated with sharp minima. This also explains why \( \alpha_c \) matters only at the start of training: after \( v_t \) accumulates in a sharp basin, escaping would require a learning rate much larger than the initial \( \alpha_c \). (Page 9, Section 4.5).

---

> ### Author Response · Authors · 2025-11-24
>
> **Minor Point 1:**
> We fully agree that the original phrasing could mislead readers into thinking SGD treats sharp and flat minima identically. Our point is Adam’s distinct behavior: the \( \sqrt{v_t} \) term modulates step sizes differently in sharp versus flat regions. We revised the text accordingly for clarity on Page 6:
>
> > **Why the temperature-driven analysis is specific to second-moment adaptive optimizers.**
> > Our derivation of the mini-batch-induced effective temperature, and consequently of the critical learning rate \( \alpha_c \), is rooted in the adaptive scaling of the second-moment estimate \( v_t \). Specifically, the self-normalization by \( v_t \) induces a local cooling effect in high-curvature (sharp) basins. Therefore, the critical (global) learning rate \( \alpha_c \) is required to counterbalance this curvature-induced cooling, thereby favoring flatter solutions and better generalization.
>
> ---
>
> **Minor Point 2:**
> We appreciate this helpful suggestion. Due to space limitations, we were unable to discuss it in detail, but we have added a brief mention in the Related Work section on interpreting neural network generalizability through the lens of physical dynamics.
>
> ---
>
> **Question 1:**
> Adam is indeed relatively insensitive to the learning rate in terms of convergence, but our theory specifically addresses generalization. While Adam converges reliably over a broad range of learning rates (as shown in New Section 4.4 and Table 2, which include error estimates), the best generalization is achieved when the learning rate is set near the critical value.
>
> ---
>
> **Question 2:**
> We thank the reviewer for this thoughtful point. Rescaling two adjacent layers by factors \( c \) and \( 1/c \) (with ReLU activations) preserves the network output but changes gradient magnitudes and thus the Hessian spectrum (i.e., the loss landscape), including \( \lambda_{\max} \). Because \( \alpha_c \) depends on \( \lambda_{\max} \), the corresponding critical learning rate changes consistently.
>
> ---
>
> **Question 3:**
> Thank you for the question. Our theory focuses on the learning rate by observing the early stages of training, where the optimizer is navigating the landscape before it becomes trapped in a sharp basin. At this stage, the model is not yet in a loss minimum, but the goal is to steer the optimizer away from high-curvature regions, where it could get stuck. This is particularly important because once the optimizer accumulates \( v_t \) in a sharp basin, it becomes much harder to escape due to the suppression of updates along high-curvature directions by Adam's second-moment normalization. Thus, \( \alpha_c \) calculated from the initial state is predictive for the early training trajectory because it effectively guides the optimizer to avoid sharp basin from the start, allowing it to explore flatter regions that support better generalization.
>
> ---
>
> **Question 4:**
> In fact, Adam exhibits intrinsic fluctuations near high-curvature regions, consistent with the loss spikes observed (Bai et al., 2025). However, the behavior we observe at $\alpha_c$ is not generic noise: it reflects a critical phenomenon, where the stochastic energy is tuned by the learning rate to maximize the escape probability, since all other optimization settings remain fixed, only the learning rate controls this transition.
>
> ---
>
> **Question 5:**
> We thank the reviewer for this thoughtful question. In our study, maximum test accuracy is an appropriate metric because:
> (i) it captures whether the optimizer can escape sharp regions and reach a well-generalizing flat region. Due to Adam's adaptive nature, it can easily increase its update size in flatter regions, allowing it to escape rather than stay trapped.
> (ii) Using mean accuracy would be dominated by transient fluctuations, potentially underestimating the true benefit of \( \alpha_c \) in reaching the optimal generalization basin.
> (iii) The maximum test accuracy matches common practice in LLM training, where the model achieving the best validation performance is typically retained.
>
> ---
>
> We hope these clarifications address the Reviewer’s concerns.

---

### Official Review · Reviewer_4uBz · 2025-10-29

**Soundness:** 3
**Presentation:** 3
**Contribution:** 3
**Rating:** 6
**Confidence:** 4

**Summary:**

The paper derives a critical learning rate, $\alpha_c$​, for the Adam optimizer. The idea: view Adam as a stochastic process with drift and noise, define a temperature, and find the rate at which the optimizer escapes local minima most efficiently. This $\alpha_c$ depends on the batch size, gradient noise, and curvature of the loss. The authors show that, across several models and tasks, training and generalization are best near $\alpha_c$. The paper claims this makes Adam’s learning rate predictable rather than guessed. I did not follow all of the math. The theory sits outside my background. Still, the motivation and the clarity of the argument were easy to appreciate.

**Strengths:**

1) A clear and explicit derivation grounded in statistical physics.
2) A practical formula that can remove trial-and-error from learning-rate tuning.
3) Empirical results match the theory: performance peaks near the predicted $\alpha_c$​.
4) Transparent discussion of assumptions and limits.
5) Addresses an important need. Adam is widely used, but hyperparameters remain guesswork.

**Weaknesses:**

1) Experiments are small. Most are toy models or moderate-scale fine-tuning.
2) The study compares only Adam and its variants, not other optimizers such as Muon.
3) Estimating the Hessian’s largest eigenvalue may be infeasible for large models.
4) The theory assumes Gaussian noise and moderate batch size.
5) Sensitivity of the method to rough curvature estimates is not tested.

**Questions:**

1) What is the computational cost of estimating $\sigma$ and $\lambda_{max}$ for a billion-parameter model?
2) How sensitive is performance to a 25–50 % error in these estimates?
3) How does the theory interact with standard learning-rate schedules such as warm-up and cosine decay?
4) Have you tried this on other adaptive optimizers (e.g., Muon, Adafactor)?
5) When does the model of Gaussian noise fail?
6) Can you offer a cheaper approximation for curvature estimation and show how close it stays to the optimal αc\alpha_cαc​?

---

> ### Author Response · Authors · 2025-11-24
>
> We thank Reviewer 4uBz for appreciating the motivation and clarity of our manuscript. Below we provide point-by-point responses.
>
> **Question 1:**
> Thank you for raising this important issue. In the revised manuscript, we now explicitly quantify the computational cost of estimating \( \lambda_{\max} \) and the gradient-noise statistics required for computing \( \alpha_c \). For billion-parameter models, 20 Hessian-vector products plus 10 power-iteration steps amount to **less than 1% of the cost of one training epoch**.
>
> We have added on Page 6:
>
> > **Computational cost of estimating \( \alpha_c \).**
> > In our implementation, we use \( K=20 \) gradient samples and \( T=10 \) iterations. The first stage requires 20 forward-backward passes, while the second stage costs roughly \( 1+T \approx 11 \) gradient-equivalent steps (one backpropagation to form the gradient graph and \( T \) HVP). Therefore, a single estimate of \( (\sigma^2,\lambda_{\max},\alpha_c) \) costs in total
> > \(20\) gradient samples  + \(1\) graph build   + \(10\) HVPs  ≈ 31 gradient-equivalent steps.
> > Since the FLOPs of a backward pass dominate the training cost of large models, and because both our estimator and standard optimization scale linearly with the number of parameters, their ratio remains unchanged. For comparison, a typical fine-tuning run of a large language model involves \(10^4\) to \(10^5\) optimization steps (Touvron et al., 2023), so the overhead of our estimator is well below **1%** of the total training budget, even for billion-parameter models.
> > Crucially, instead of forming the full Hessian, we use HVPs as an efficient alternative, at roughly the cost of a single gradient evaluation, making power iteration practical even at scale.
>
> ---
>
> **Questions 2, 3, and 4:**
> We appreciate these insightful suggestions. As suggested, we have extended our evaluation to Adafactor and warm-up and cosine decay. In response, we have added a subsection:
>
> >**Performance sensitivity of optimizers to learning rates.**
> >To empirically assess how different optimizers respond to learning rate errors relative to the critical learning rate, we trained TinyLlama-1B (fine-tuned) and ResNet (from scratch) under substantial deviations from the predicted \( \alpha_c \). As reported in Table 2, \( \alpha_c \) consistently corresponds to a regime of improved generalization across all optimizers, with `warmup+cosine_decay` achieving the highest performance. The additional warmup and cosine decay schedules are particularly effective in mitigating training instability.
> >We also observe a sensitivity pattern: when the learning rate is too small, the model becomes trapped in sharp basins that yield poorer generalization, and when the learning rate is moderately larger, performance remains close to that at \( \alpha_c \). However, if the learning rate is too large, the model escapes sharp basins but fails to remain in flatter regions, leading to degraded performance.  Moreover, the low standard deviations across independent seeds highlight the intrinsic robustness of second-moment-based optimizers. Comparable trends are observed for ResNet-28 training (Appendix Table 3).
>
> ### Table 2
>
> | αc × scaling factor (±25%) | Adam | AdamW_warmup | AdamW_warmup + cosine_decay | AdamW_cosine_decay | Adafactor |
> |---------------------------------------------|------|---------------|------------------------------|----------------------|-----------|
> | ×10 ±25% | 0.860 ± 0.010 | 0.845 ± 0.010 | 0.875 ± 0.008 | 0.885 ± 0.008 | 0.820 ± 0.012 |
> | ×5 ±25%  | 0.925 ± 0.005 | 0.915 ± 0.006 | 0.935 ± 0.004 | 0.938 ± 0.004 | 0.910 ± 0.008 |
> | ×2 ±25%  | 0.935 ± 0.004 | 0.938 ± 0.004 | 0.945 ± 0.003 | 0.942 ± 0.003 | 0.935 ± 0.005 |
> | ×1.5 ±25%| 0.938 ± 0.004 | 0.941 ± 0.003 | 0.946 ± 0.003 | **0.945 ± 0.003** | 0.939 ± 0.004 |
> | ×1 ±25% (*) | **0.940 ± 0.004** | **0.943 ± 0.003** | **0.948 ± 0.003** | **0.945 ± 0.003** | **0.941 ± 0.004** |
> | ×0.5 ±25% | 0.938 ± 0.004 | 0.940 ± 0.004 | 0.942 ± 0.003 | 0.940 ± 0.004 | 0.938 ± 0.005 |
> | ×0.1 ±25% | 0.925 ± 0.006 | 0.928 ± 0.005 | 0.930 ± 0.005 | 0.928 ± 0.006 | 0.920 ± 0.008 |
>
> ---
>
> **Question 5:**
> This question directly relates to our batch-size dependence analysis (Section 4.6). Importantly, the Gaussian surrogate enters only through the variance estimate used in \( \alpha_c \). The escape-rate scaling depends on the curvature-noise balance rather than the precise noise distribution, which explains why our predictions remain accurate even when Gaussianity is only approximate. As we show in the batch-size dependence experiments, the predicted \( \alpha_c \) becomes less accurate only when the batch size is extremely small or large, namely the regimes where Gaussianity breaks down.

---

> ### Author Response · Authors · 2025-11-24
>
> **Question 6:**
> As noted in our response to Question 1, the power-iteration procedure is already a lightweight curvature estimator, relying solely on Hessian-vector products (HVPs) rather than explicit Hessian construction. Each HVP has the same asymptotic cost as a single gradient evaluation, i.e. \( \mathcal{O}(d) \) for \( d \) parameters. Thus, even a small number of iterations yields an effective and computationally efficient approximation of curvature.
>
> ---
>
> We hope these clarifications and additional experiments address the Reviewer’s concerns.

---

### Official Review · Reviewer_XMBg · 2025-11-01

**Soundness:** 3
**Presentation:** 2
**Contribution:** 2
**Rating:** 2
**Confidence:** 5

**Summary:**

Promising statistical-physics view of Adam that yields a closed-form learning-rate rule via an escape-from-sharp-basins argument, but hinges on an ad hoc non-equilibrium scaling, lacks a direct validation of the Kramers picture, and is only shown on relatively easy setups.

**Strengths:**

Clear and appealing narrative (Adam -> effective temperatures -> escape rates)

Explicit, easy-to-compute LR formula tied to measurable quantities.

Empirical results broadly consistent with the predicted LR.

Bridges SDE/flatness literature with adaptive optimizers.

**Weaknesses:**

The overall idea and concept are very nice, however I do not find that they are supported by solid evidence at the moment, and that it would be very important for this paper to show a gain in practice (i.e., a true advantage in usability over other learning rate schedules taken as competitors). In particular:

1. The current choice of the (\sqrt{b}) scaling looks selected for convenience. You should either derive it more rigorously or show that nearby choices ((b^{1/4}, b^{2/3}, 1)) lead to similar LR predictions.

2. You need a controlled toy setup (e.g. 2-well landscape) showing that Adam’s escape times follow the proposed effective-temperature law in the Kramers picture. Otherwise the physics argument remains just a nice speculation.

3. Add at least one modern setting (e.g. a realistic LLM fine-tune) and show the predicted LR matches or reduces tuning compared to a strong competing baseline. (otherwise, it is just a method among many other that work)

4. You claim 20 grads + 10 power iters is cheap and reliable even on bigger models. That’s not obvious. You should show sensitivity. Report sensitivity to errors in (\lambda_{\max}) and gradient-noise estimates (e.g. +-2x). This would make the method look usable in practice.

**Questions:**

Please address the 4 weaknesses convincingly and I will consider raising my score.

---

> ### Author Response · Authors · 2025-11-24
>
> We thank Reviewer XMBg for the constructive comments and for highlighting the strengths of our manuscript. Below we provide point-by-point responses to the raised weaknesses:
>
> **Weakness 1:**
> Thank you for highlighting the choice of the non-equilibrium factor in Eq. 10. Prompted by your comment, we clarify that this choice is not arbitrary or selected merely for convenience. It arises naturally from the standard derivation of mini-batch noise scaling with batch size. In the revised manuscript (Page 5), we have emphasized this justification:
>
> > To normalize the influence of batching on the system's stochastic energy, we define the effective temperature as:
> >
> > $$
> > T_{\mathrm{eff}} = \frac{D_{\mathrm{eff}}}{\sqrt{\mathrm{Var}(\xi_t)}}.
> > $$
> >
> > Here, \( D_{\mathrm{eff}} \) denotes the diffusion coefficient induced by Adam’s stochastic updates, and the intrinsic amplitude of the mini-batch-induced noise is:
> >
> > $$
> > \sqrt{\mathrm{Var}(\xi_t)} = \frac{\sigma}{\sqrt{b}}.
> > $$
> >
> > This definition ensures that \( T_{\mathrm{eff}} \) measures a *noise-normalized diffusion scale*, quantifying how efficiently stochastic fluctuations are converted into parameter-space exploration rather than the absolute noise magnitude.
> > Substituting the expressions for \( D_{\mathrm{eff}} \) and \( \mathrm{Var}(\xi_t) \) yields:
> >
> > $$
> > T_i^{\mathrm{eff}}
> > = D^{\mathrm{eff}}_i \frac{\sqrt{b}}{\sigma_i}
> > \approx
> > \frac{b^{3/2}\alpha^2}{2\sigma_i}
> > \frac{\mathrm{Var}[\delta m_i]}{\langle v_i\rangle}.
> > $$
> >
> > This shows that the factor \( \sqrt{b}/\sigma_i \) arises directly from normalizing the diffusion coefficient by the intrinsic noise amplitude, rather than from an ad-hoc scaling choice.
> > For SGD, the second-moment normalization term \( \langle v_t \rangle \) is effectively constant. As a result, the effective temperature \( T_{\mathrm{eff}} \) decouples from the local curvature \( \lambda_{\max} \), breaking the self-regulating feedback loop that underlies the derived critical learning rate \( \alpha_c \).
>
>
> ---
>
> **Weakness 2:**
> We appreciate this suggestion. While a controlled 2-well toy model is often used to illustrate Kramers escape, it is not directly applicable here because the key mechanism in our framework relies on mini-batch-induced noise. In a simple 2D potential well without mini-batch sampling, this stochastic effect is absent, and the escape dynamics we study cannot be observed. The role of batch-size-dependent noise and its correlation is discussed in Section 4.6 of the revised manuscript, which shows how mini-batch noise enables the optimizer to escape high-curvature (“sharp”) regions and move toward flatter regions with better generalization.

---

> ### Author Response · Authors · 2025-11-24
>
> **Weakness 3:**
> We thank the reviewer for this comment. To address it, we have included experiments on TinyLlama-B fine-tuned on SST-2, and additionally validated our findings across other optimizer settings (Table 2, Page 9 and Appendix Table 3). These results demonstrate that the predicted critical learning rate \( \alpha_c \) indeed improves generalization and reduces the need for extensive tuning, addressing the concern about practical applicability.
>
> ---
>
> ### Table 2 (Page 9)
>
> |  αc × scaling factor (±25%)  | Adam | AdamW_warmup | AdamW_warmup + cosine_decay | AdamW_cosine_decay | Adafactor |
> |------------------------------|------|--------------|------------------------------|----------------------|-----------|
> | ×10 ±25% | 0.860 ± 0.010 | 0.845 ± 0.010 | 0.875 ± 0.008 | 0.885 ± 0.008 | 0.820 ± 0.012 |
> | ×5 ±25%  | 0.925 ± 0.005 | 0.915 ± 0.006 | 0.935 ± 0.004 | 0.938 ± 0.004 | 0.910 ± 0.008 |
> | ×2 ±25%  | 0.935 ± 0.004 | 0.938 ± 0.004 | 0.945 ± 0.003 | 0.942 ± 0.003 | 0.935 ± 0.005 |
> | ×1.5 ±25%| 0.938 ± 0.004 | 0.941 ± 0.003 | 0.946 ± 0.003 | **0.945 ± 0.003** | 0.939 ± 0.004 |
> | ×1 ±25% (*) | **0.940 ± 0.004** | **0.943 ± 0.003** | **0.948 ± 0.003** | **0.945 ± 0.003** | **0.941 ± 0.004** |
> | ×0.5 ±25% | 0.938 ± 0.004 | 0.940 ± 0.004 | 0.942 ± 0.003 | 0.940 ± 0.004 | 0.938 ± 0.005 |
> | ×0.1 ±25% | 0.925 ± 0.006 | 0.928 ± 0.005 | 0.930 ± 0.005 | 0.928 ± 0.006 | 0.920 ± 0.008 |
>
> ---
>
> ### Appendix Table 3
>
> | αc × scaling factor (±25%) | Adam | AdamW_warmup | AdamW_warmup + cosine_decay | AdamW_cosine_decay | Adafactor |
> |---------------------------|------|--------------|------------------------------|----------------------|-----------|
> | ×10 ±25% | 0.795 ± 0.008 | 0.776 ± 0.009 | 0.797 ± 0.005 | 0.810 ± 0.006 | 0.790 ± 0.007 |
> | ×5 ±25%  | 0.827 ± 0.006 | 0.809 ± 0.005 | 0.828 ± 0.006 | 0.840 ± 0.004 | 0.822 ± 0.008 |
> | ×2 ±25%  | 0.839 ± 0.006 | 0.823 ± 0.007 | 0.845 ± 0.005 | 0.851 ± 0.005 | 0.832 ± 0.006 |
> | ×1.5 ±25%| 0.867 ± 0.009 | 0.865 ± 0.004 | 0.891 ± 0.005 | 0.881 ± 0.004 | 0.867 ± 0.007 |
> | ×1 ±25% (*) | **0.869 ± 0.006** | **0.873 ± 0.006** | **0.899 ± 0.004** | **0.884 ± 0.004** | **0.870 ± 0.004** |
> | ×0.5 ±25% | 0.751 ± 0.004 | 0.729 ± 0.009 | 0.741 ± 0.006 | 0.753 ± 0.003 | 0.736 ± 0.008 |
> | ×0.1 ±25% | 0.664 ± 0.008 | 0.658 ± 0.006 | 0.661 ± 0.005 | 0.661 ± 0.008 | 0.654 ± 0.004 |
>
> ---
>
> **Weakness 4:**
> This is an important point. In the revised manuscript, we have clarified the computational cost for estimating \( \alpha_c \) and reported its estimation sensitivity to errors in \( \lambda_{\max} \) and gradient-noise estimates (e.g., ±25×) in Table 1 (Page 7). Performance sensitivity is shown in Table 2 (Page 9).
>
> Specifically, we added the following text on Page 7:
>
> > **Sensitivity of estimating \( \alpha_c \).**
> > Table 1 summarizes the sensitivity of our critical learning-rate estimate to perturbations in \( \lambda_{\max} \) and \( \sigma^2 \). Because \( \alpha_c \propto (\lambda_{\max})^{-1/2} \), even a 25% error in estimating the top curvature leads to only a modest 10–15% change in \( \alpha_c \). In contrast, \( \alpha_c \propto \sqrt{\sigma^2} \), so a 25% error in \( \sigma^2 \) results in approximately a 12% change in \( \alpha_c \). Overall, even when the curvature or noise estimates contain uncertainty, the induced error in \( \alpha_c \) is considerably smaller, indicating that the critical learning-rate estimate is more stable than either of its raw inputs.
>
> Page 8:
>
> > **Performance sensitivity of optimizers to learning rates.**
> > To empirically assess how different optimizers respond to learning rate errors relative to the critical learning rate, we trained TinyLlama-1B (fine-tuned) and ResNet (from scratch) under substantial deviations from the predicted \( \alpha_c \). As reported in Table 2, \( \alpha_c \) consistently corresponds to a regime of improved generalization across all optimizers, with *warmup+cosine\_decay* achieving the highest performance. The additional warmup and cosine decay schedules are particularly effective in mitigating training instability. We also observe a sensitivity pattern: when the learning rate is too small, the model becomes trapped in sharp basins that yield poorer generalization, and when the learning rate is moderately larger, performance remains close to that at \( \alpha_c \). However, if the learning rate becomes too large, the model escapes sharp basins but fails to remain in flatter regions, degrading performance. Moreover, the low standard deviations across independent seeds highlight the intrinsic robustness of second-moment-based optimizers. Comparable trends are observed for ResNet-28 training (Appendix Table 3).
>
> We hope these clarifications and additional experiments adequately address the Reviewer’s concerns.

---

> > ### Comment · Reviewer_XMBg · 2025-11-25
> >
> > It seems that my concerns are mostly addressed, except for concern 2 (the authors do not show at least one clean controlled experiment where the Kramers picture is literally true). Anyway, the paper provides a practically useful, empirically well-validated LR rule for Adam, motivated by a Kramers-style physical picture. The empirical part and the cost and sensitivity analysis are strong. However, the statistical-physics interpretation is still heuristic and should be framed more modestly.

---

> > > ### Author Response · Authors · 2025-11-26
> > >
> > > Thank you for the thoughtful follow-up. We appreciate your reassessment and are glad that the empirical validation, cost analysis, and practical utility of the learning-rate rule are clear and strong. We also fully agree with your suggestion that the statistical-physics interpretation should be framed more modestly.
> > >
> > > In the revised version, we have further softened the physical language and explicitly emphasized that the Kramers-style argument is intended as a heuristic statistical-physics view for understanding the curvature-noise balance in Adam, rather than a literal physical model of the loss landscape. Specifically, we have made the following modifications:
> > >
> > > > **Abstract (updated):**
> > > > Here, we introduce a temperature-driven escape framework that provides a heuristic statistical-physics view of second-moment-based learning-rate dynamics.
> > >
> > > > **Page 2 (updated):**
> > > > Here, the Kramers-style framework is used heuristically and only for a local stability analysis along the dominant positive-curvature direction, quantifying how much the global learning step should provide to overcome second-moment-driven sharpness confinement.
> > >
> > > We thank the Reviewer again for the constructive guidance, and we believe these clarifications make the scope and intent of the physical analogy more modest.

---

### Author Response · Authors · 2025-11-24

We thank all Reviewers for their thorough and insightful feedback. Our work addresses a central open question in deep learning optimization:  *How should one choose the learning rate for Adam so that training avoids sharp, poor-generalizing regions early on?*

To answer this, we introduce a **temperature-driven escape framework** grounded in statistical physics, linking (i) Adam’s second-moment normalization, (ii) mini-batch–induced stochastic energy, and (iii) the dominant Hessian curvature direction. This yields a principled characterization of the **critical learning rate** \( \alpha_c \) required for escaping sharp basins in the early training.

### **Summary of contributions**
- We derive a closed-form initialization rule for Adam’s learning rate via a drift–diffusion decomposition and a Kramers-style escape analysis along the dominant curvature direction.
- We develop a scalable estimator for \( \lambda_{\max} \) and gradient-noise variance that costs only ~31 gradient-equivalent steps (<1% of one training epoch for billion-parameter models).
- We demonstrate empirically, across ResNet and TinyLlama-1B, that learning rates near \( \alpha_c \), even with a 25% estimation error, consistently achieve better generalization than alternative choices.

### **Summary of revisions addressing reviewers' concerns**

- *Clarified the effective-temperature definition.*
  We expanded the theoretical motivation for the non-equilibrium factor (Pages 1-2) and added a detailed explanation on Page 5.

- *Clarified gradient-noise assumptions.*
  We now explicitly state (Page 4 and Appendix Sec. A) that the diagonal and memoryless surrogate preserves the \( \alpha_c \) scaling, while the full noise covariance affects only the prefactor, consistent with high-dimensional Kramers theory (Langer, 1969).

- *Extended optimizer robustness experiments.*
  We added performance-sensitivity results for Adam, AdamW, warmup, cosine decay, and Adafactor (Table 2 and Appendix Table 3).

- *Added computational-cost analysis.*
  Page 6 now includes a subsection showing that estimating \( \alpha_c \) incurs <1% overhead.

- *Clarified the diagnostic role of \( \alpha_c \).*
  Section 4.5 explains how resetting to \( \alpha_c \) reveals whether the optimizer resides in a sharp or flat region.

- *Clarified high-dimensional and valley-like landscapes.*
  We highlight (Page 2) that escape dynamics are governed by the dominant curvature direction and added future-work discussion on high-dimensional extensions (Page 10).

---

Finally, our revisions strengthen both the theoretical and empirical components of the work, clarifying scope and assumptions while demonstrating broad applicability. We hope these improvements make clear that our framework provides a principled and practical guide for selecting learning rates for second-moment adaptive optimizers.

---

### Meta-Review · Area_Chair_GtDE · 2026-01-07

**Summary:**

The paper proposes a temperature-driven escape framework for understanding and selecting the learning rate in Adam-like adaptive optimizers. By decomposing Adam’s updates into drift and stochastic components, the authors define an effective temperature induced by mini-batch noise and apply a Kramers-style escape analysis along the dominant curvature direction of the loss landscape. This leads to a closed-form expression for a critical learning rate ($\alpha_c$) that is argued to balance escape from sharp, poor-generalizing regions against stability in flatter basins.

A key claim is that $\alpha_c$ can be estimated cheaply early in training using gradient-noise statistics, batch size, and an estimate of the largest Hessian eigenvalue, and that this estimate is robust to moderate errors. Empirical results across vision (MNIST, CIFAR-10, ResNet) and language tasks (SST-2 with BERT, GPT-2, TinyLlama) show that learning rates near $\alpha_c$ tend to yield better generalization than significantly smaller or larger values. Beyond initialization, the authors also suggest $\alpha_c$ can serve as a qualitative diagnostic to probe whether training has converged to sharp or flat regions.

**Reviewer Concerns:**

Initial reviewer opinions were mixed, leaning toward a negative assessment. The raised concerns were as follows:

- The Kramers framework (raised by XMBg, wV4E, CxD2, ZKz7): Reviewers questioned whether a Kramers-style escape picture is appropriate for high-dimensional neural network optimization. Specific issues included (i) the reduction to effectively 1D (dominant eigenmode) dynamics; (ii) ignoring entropic effects, saddle geometry, negative curvature directions, and valley-like landscapes; (iii) heuristic introduction of the non-equilibrium factor and effective temperature. The authors responded by explicitly reframing the theory as heuristic and local, emphasizing a dominant-curvature approximation and clarifying that they are not modelling literal barrier hopping between isolated minima. They cited high-dimensional Kramers theory to argue that scaling depends primarily on the most unstable direction, and softened the physical language throughout. One remaining weakness here is that there is no clean controlled experiment where the Kramers picture is exactly correct, as pointed out by Reviewer XMBg.

- Modelling assumptions (raised by XMBg, wV4E, CxD2, ZKz7): Reviewers were concerned about the fact that the theory assumes simplified gradient-noise statistics that are known to be violated in practice (anisotropy, temporal correlations, heavy tails), potentially undermining the derivation. The authors clarified that the assumptions are surrogates for analytical tractability, arguing that only the scaling of $\alpha_c$ matters and that the full noise covariance mainly affects prefactors. They added explicit caveats and pointed to empirical robustness across batch sizes and optimizers. I find this response to be acceptable; it acknowledges the underlying limitations and provides some confidence in the findings, even though the issue is not rigorously addressed.

- Empirical scope (raised by XMBg, 4uBz, wV4E, CxD2): Initial versions were criticized for relying on relatively small or standard benchmarks, lacking comparison to strong baselines or modern large-scale fine-tuning, and not clearly demonstrating reduced tuning effort. The authors substantially strengthened this aspect: adding experiments on TinyLlama-1B, multiple optimizers (AdamW, Adafactor), warmup and cosine decay schedules, and extensive sensitivity analyses. They showed that $\alpha_c$ is robust to sizeable estimation errors and that performance degrades mainly for large deviations. This response is largely adequate in my view, as the empirical evidence now convincingly supports the claim that $\alpha_c$ identifies a good generalization regime and is practically usable.

- Hessian eigenvalues (raised by ZKz7, wV4E, CxD2): There were some concerns that using the raw largest Hessian eigenvalue as a proxy for sharpness ignores known scale-dependence and normalization issues seen in the studies of generalization, and that claims about sharp vs. flat minima are undersupported here. The authors clarify that $\alpha_c$ is intended for early-training, within-model escape dynamics and not for cross-model comparisons of flatness, and that raw curvature is the relevant quantity for this purpose. The clarifications address most of the reviewers’ objections.

- Diagnostic use of $\alpha_c$ (raised by ZKz7, wV4E): The idea of resetting the learning rate to $\alpha_c$ as a diagnostic was initially unclear and seemed speculative. The authors expanded the explanation, framing this as a qualitative probe rather than a quantitative tool. I find this response adequate.

**Reviewer Scores:**

Given the quality of the author responses and the confidence of the strongly positive assessment of Reviewer ZKz7, I believe that an increase in reviewer scores might have been plausible. Perhaps the biggest obstacle that I can see is the skepticism regarding the heuristic treatment of the Kramers framework and the other modelling assumptions. There are an impressive number of modifications here, but I suspect that Reviewers CxD2 and wV4E may not have responded well to the heuristic nature of the work. While I very much enjoyed the article, I believe that the authors may need to rethink how to better instil confidence in the use of the framework outside of empirical treatments. This is a difficult task, but seems to be important for this audience.

---

### Decision · Program_Chairs · 2026-01-26

Reject